# Future hydrogen economies imply environmental trade-offs and a supply-demand mismatch

Tom Terlouw [1,2,3] ✉, Lorenzo Rosa [4], Christian Bauer [3] & Russell McKenna [2,5] ✉

Hydrogen will play a key role in decarbonizing economies. Here, we quantify the costs and environmental impacts of possible large-scale hydrogen economies, using four prospective hydrogen demand scenarios for 2050 ranging from 111–614 megatonne $H_2$ year$^{-1}$. Our findings confirm that renewable (solar photovoltaic and wind) electrolytic hydrogen production generates at least 50–90% fewer greenhouse gas emissions than fossil-fuel-based counterparts without carbon capture and storage. However, electrolytic hydrogen production could still result in considerable environmental burdens, which requires reassessing the concept of green hydrogen. Our global analysis highlights a few salient points: (i) a mismatch between economical hydrogen production and hydrogen demand across continents seems likely; (ii) region-specific limitations are inevitable since possibly more than 60% of large hydrogen production potentials are concentrated in water-scarce regions; and (iii) upscaling electrolytic hydrogen production could be limited by renewable power generation and natural resource potentials.

Future climate mitigation scenarios highlight massive hydrogen requirements, accounting for 2–10% of global final energy consumption by 2050[1–4]. Meeting such demand requires an upscaling of hydrogen production from the current 90 million metric tonnes (Mt) per year to 200–600 Mt year$^{-1}$[1,5]. Hydrogen is an energy carrier that is supposed to play a key role in decarbonizing hard-to-electrify sectors[6,7], storing energy from intermittent renewable electricity sources[8], and is needed as chemical feedstock as well as precursor of synthetic hydrogen-based hydrocarbons, so-called e-fuels[9,10].

Hydrogen production relies on carbon-intensive methods, particularly steam methane reforming of natural gas and coal gasification[6]. Consequently, hydrogen production contributes to approximately 2% of energy-related global carbon dioxide emissions[5], necessitating the need for a transition towards green (i.e., low-carbon) hydrogen production. However, even very low-carbon green hydrogen causes

indirect emissions and material flows that require assessing environmental trade-offs and potential bottlenecks for large-scale deployment[8]. Furthermore, how and where to produce these massive amounts of green hydrogen remains largely unknown. Overall, comprehensive assessments are missing that consider: (i) the entire portfolio of possible hydrogen production technologies—such as biomass-based hydrogen production and methods including carbon capture and storage—in addition to water electrolysis; (ii) more than one future global hydrogen scenario; and (iii) both cost and a comprehensive set of environmental burdens including potential resource constraints. Some recent studies[8,11–14] show potential implications of hydrogen production in certain geographical locations or case studies. Other studies are limited in their scope due to a focus on hydrogen production by water electrolysis[8,15], biomass[16], impacts on land and water scarcity[15,17], and one socio-economic development scenario

[1]Separation Processes Laboratory, Institute of Energy and Process Engineering, ETH Zurich, Zurich 8092, Switzerland. [2]Chair of Energy Systems Analysis, Institute of Energy and Process Engineering, ETH Zurich, Zurich 8092, Switzerland. [3]Technology Assessment Group, Laboratory for Energy Systems Analysis, 5232 Villigen PSI, Switzerland. [4]Department of Global Ecology, Carnegie Institution for Science, Stanford, CA 94035, USA. [5]Laboratory for Energy Systems Analysis, 5232 Villigen PSI, Switzerland. ✉e-mail: tom.terlouw@psi.ch; russell.mckenna@psi.ch

only[8,15]. These omissions impede a comprehensive understanding of the potential challenges and opportunities associated with the transition towards a hydrogen-based economy.

To provide such a comprehensive understanding, we conduct a global geospatial analysis with specific yields of renewables—using onshore and offshore wind as well as residential and utility solar photovoltaic (PV)—and integrate techno-economic assessment and (prospective) environmental life cycle assessment (LCA). This allows the identification of trade-offs between different objectives for upscaling hydrogen production on a global level and differentiates between more and less suitable regions considering different future hydrogen economies in 2050. We show where large quantities of hydrogen should be produced in the future from an economic and environmental perspective, considering cost and environmental constraints, such as water, land, and materials. Finally, cost- and environmental trade-offs are determined on a global scale.

Here, we show several significant insights: (i) While green hydrogen emits 50–90% less greenhouse gas (GHG) emissions compared to steam methane reforming, it can still result in considerable GHG emissions and environmental burdens for a large-scale hydrogen economy due to embodied emissions of energy technologies and global warming potential of hydrogen leakage; (ii) there exists a mismatch between economical hydrogen production and hydrogen demand across continents; (iii) regional-specific limitations emerge since large hydrogen production potentials are concentrated in water-scarce regions; and (iv) the upscaling of hydrogen production might be constrained by renewable electricity requirements and natural resources.

## Results
Future climate mitigation scenarios (Fig. 1g) illustrate that electrolytic hydrogen production is the most dominant future low-carbon hydrogen production pathway in ambitious climate scenarios[5,18]. Thus, our focus is on electrolytic hydrogen production via polymer electrolyte membrane (PEM) electrolyzers, to complement other hydrogen production pathways based on natural gas, coal, and biomass (Fig. 1g). The reference year and four different scenarios are included to consider a global scale-up of hydrogen production, based on a current scenario (2022) and four future scenarios valid for 2050: Reference (2022), Business-as-usual (2050), 2 °C (2050), 1.5 °C, 1.5 °C – IRENA. The scenarios differ in terms of cost assumptions and emission factors (using *premise*[19]), as described in the Methods section.

### Environmental burdens of hydrogen production pathways
Recent techno-economic and environmental life cycle analyses[8,13,14,19] reveal the impacts of location-specific conditions on hydrogen production. Data from these studies shown the environmental burdens of hydrogen production now and in the future (2050) as provided in Supplementary Figs. 8, 10, and 11.

Our results show that the capacity factor of renewables is decisive for achieving low-carbon hydrogen production via water electrolysis. With low capacity factors of renewables, climate change impacts can exceed the green hydrogen production level (e.g., as set by CertifHy of 4.4 kgCO$_2$-eq. kg$^{-1}$H$_2$)[20] due to embodied emissions and hydrogen leakage. Hydrogen production via water electrolysis using solar PV electricity might result in substantial emissions in the short term—due to current PV wafer production in China with fossil energy—if the capacity factor of the solar PV system is low (resulting up to 5.6 kgCO$_2$-eq. kg$^{-1}$H$_2$). However, it can be reduced using ground-mounted solar PV systems as they have smaller embodied emissions from production (2.6–4.5 kgCO$_2$-eq. kg$^{-1}$H$_2$). In contrast, hydrogen production via water electrolysis with wind turbines has substantially lower emissions (1.1–1.8 kgCO$_2$-eq. kg$^{-1}$H$_2$) due to a higher capacity factors and lower embodied emissions per unit of electricity needed for the electrolyzer. Our future scenarios reveal, however, that emissions from water

electrolysis are likely to be reduced due to improved technology efficiencies and the overall decarbonization of the energy system. Modifying the electricity, cement, steel, and fuels sector in the background LCA database (using *premise*[19]) results in a reduction to 0.8–1.4 kgCO$_2$-eq. kg$^{-1}$H$_2$ and to 0.4–0.8 kgCO$_2$-eq. kg$^{-1}$H$_2$ for hydrogen production via water electrolysis using solar PV and wind energy sources, respectively.

This highlights that using a global single life cycle emission factor —e.g., per unit of energy—for renewables is inappropriate to capture environmental burdens in a generic way. Such an approach fails to adequately account for location-specific environmental burdens, making it unsuitable for geospatial analyses with a global scope. Second, electrolytic hydrogen production pathways generally exhibit significantly lower GHG emissions (50–90%) compared to pathways relying on natural gas without carbon capture and storage. Third, applying low-carbon future energy scenarios in LCA, both in the foreground and background[19], leads to substantially lower GHG emissions for renewable-based hydrogen production routes (less than 1.4 kgCO$_2$-eq. kg$^{-1}$H$_2$). Each hydrogen production pathway entails some environmental trade-offs (Supplementary Fig. 9), although hydrogen production with highly abundant wind energy sources seems to be the most favorable option from an environmental perspective. These findings confirm the urgent need to assess life cycle impacts when evaluating hydrogen production pathways, emphasizing the need to consider factors beyond the operational phase, such as embodied emissions.

### Location-specific impacts
We focus on wind-based (onshore and offshore) and solar PV-based electrolytic hydrogen production in Fig. 2, i.e., potential global electrolytic hydrogen production locations. The left column demonstrates specific global hydrogen production cost in each grid cell—from 1–5 € kg$^{-1}$ H$_2$—for the reference situation and three future scenarios considered. The right column illustrates the specific life cycle GHG emissions from electrolytic-based hydrogen production (from 0–4.4 kg CO$_2$-eq. kg$^{-1}$ H$_2$).

This figure highlights that a cost and GHG emission reduction of electrolytic hydrogen production highly depends on the future socio-economic narrative; the more ambitious the climate scenario, the more cost reductions and technological improvements are expected for renewables and electrolyzers. Electrolytic hydrogen production is currently expensive compared to steam methane reforming (SMR), and only specific geographical regions exhibit relatively low (3 € kg$^{-1}$ H$_2$) electrolytic hydrogen production costs, such as coastal zones in North-Western Europe, Canada, and the center of the USA.

The more ambitious climate scenarios indicate that electrolytic hydrogen production costs of less than 2 € kg$^{-1}$ H$_2$ are reachable for large geographical regions, mainly areas in Australia, USA, Canada, the North-West of Europe, and the Sahara. Similar results in terms of best-performing regions are obtained concerning GHG emissions.

### A mismatch between economical supply and demand
Economical locations are used to determine potential future global hydrogen production hubs while continental hydrogen demand data is based on The REgional Model of INvestments and Development (REMIND) in 2050[18,21]; economical hydrogen production and continental hydrogen demand are illustrated in Fig. 3. The four subplots in the first column illustrate the cost supply curve of hydrogen (on the y-axis, to the left bottom of the subplots) up to the total hydrogen production potential, considering the land use utilization factors described in the Methods section. Vertical dashed lines in dark blue and black represent hydrogen production from water electrolysis and overall hydrogen demand in the analyzed scenarios (based on the outputs of REMIND[18,21]), respectively.

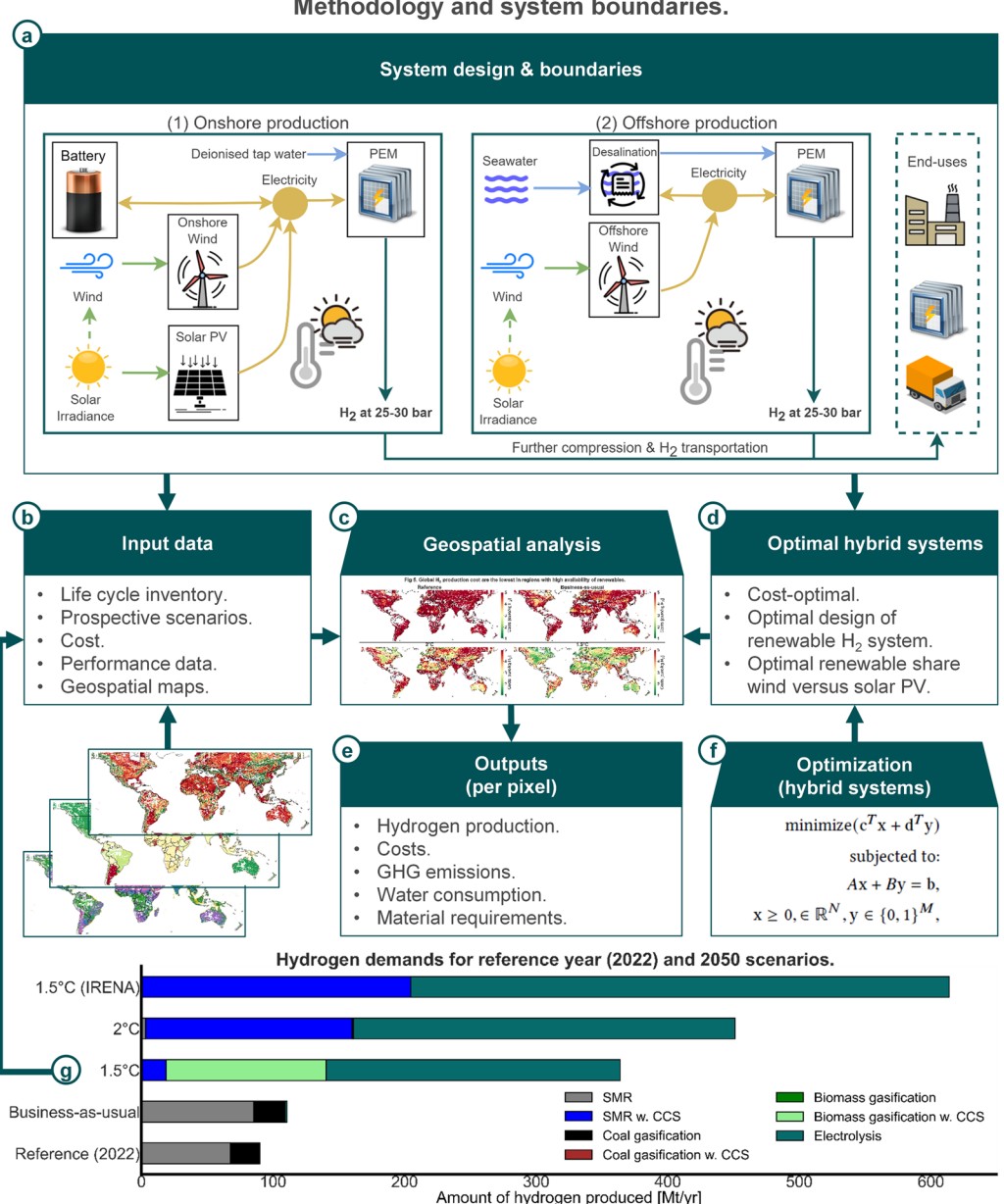

**Fig. 1 | Graphical overview of the methodology. a** System boundaries and hydrogen production configurations. **b** Input data. **c** Geospatial analysis (method). **d** Data from optimal hybrid energy systems. **e** Output data from geospatial analysis. **f** Optimization (method) of hybrid energy systems. **g** Hydrogen demand scenarios with production technologies. PEM polymer electrolyte membrane, PV photovoltaic, SMR steam methane reforming, w. CCS with carbon capture and storage, IRENA International Renewable Energy Agency, GHG greenhouse gas.

The horizontal dashed red line serves as a reference to show the current cost of fossil-fuel-based hydrogen production−using historic natural gas prices of 23 € MWh$^{-1}$−including a potential carbon price of 100 € t$^{-1}$CO$_2$, using the carbon footprint of steam methane reforming as a reference. The subplots in the second column to the right highlight the selected economical geographical regions to meet electrolytic hydrogen demand.

The two subplots at the bottom of Fig. 3 show the geographical mismatch between hydrogen production and demand based on the 2 °C (451 Mt H$_2$ year$^{-1}$) and 1.5 °C (364 Mt H$_2$ year$^{-1}$) scenarios from REMIND. Dark green colors illustrate an oversupply of hydrogen while dark red colors indicate shortages of hydrogen supply indicating import demand. India and South-East Asia have a deficit of low-cost hydrogen production locations due to their less suitable climate, land availability, and high population densities. On the contrary, Canada, Africa, the USA, and Australia are the only continents with sufficient

economic hydrogen production in both scenarios, implying that the conversion to hydrogen (carriers) and transportation is inevitable for a future low-carbon economy if large-scale hydrogen economies are established. Importantly, this requires the development of a dedicated hydrogen transportation network, which has not been analyzed in this work but would require additional investments and lead to additional impacts (Discussion).

Applying a carbon price−the black dashed line in Fig. 3 representing 100 € t$^{-1}$CO$_2$−in ambitious decarbonization scenarios results in cost-competitivity with SMR for most electrolytic-based hydrogen production in 1.5 °C scenarios.

These future scenarios illustrate the highest economical hydrogen production potential in Australia, parts of the Sahara desert, Canada, the North-West of Europe, the USA, and the North of China. It is worth noting that the capacity factors of solar PV and wind energy sources highly influence these results.

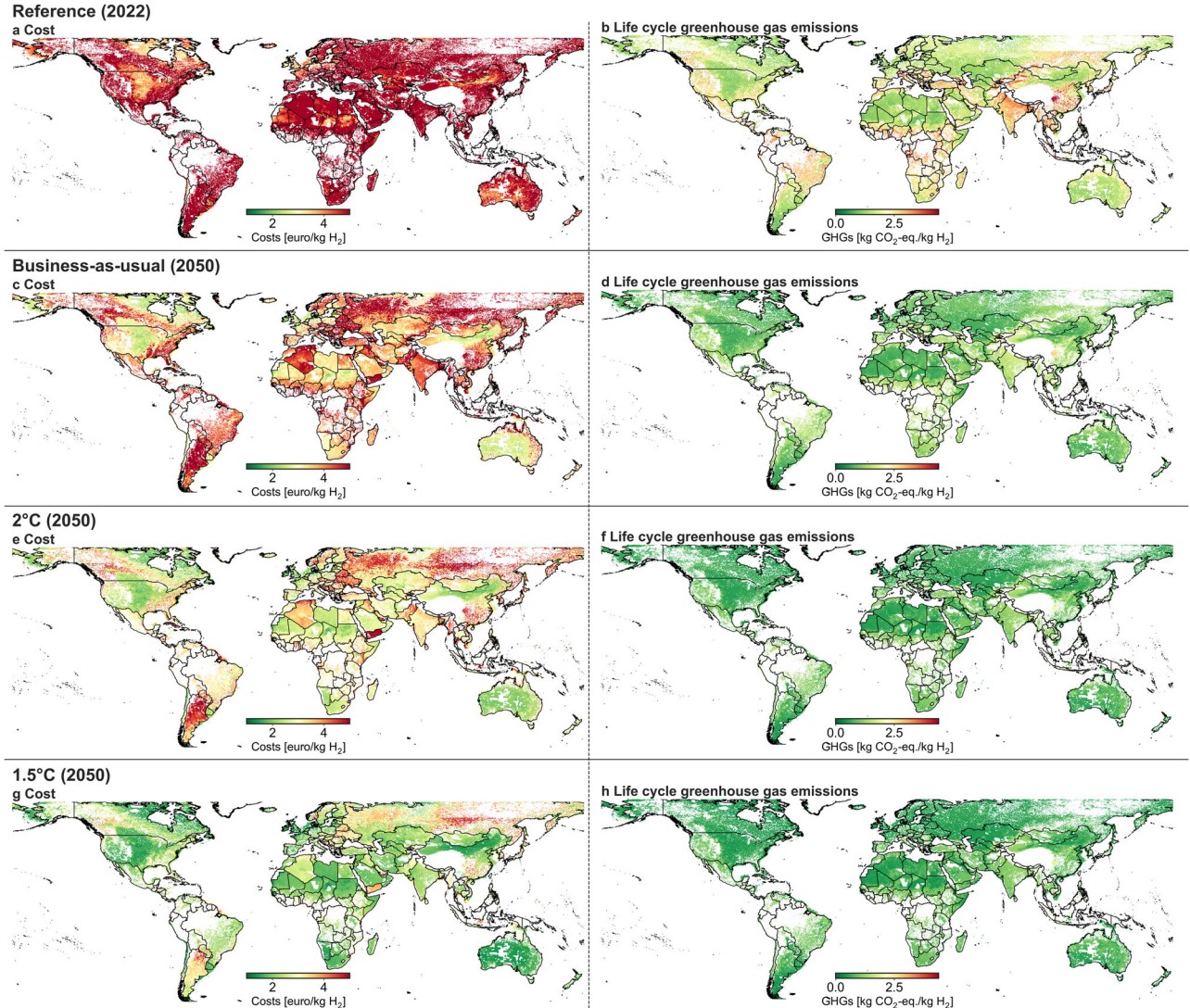

**Fig. 2 | Cost and GHG emissions are highly location-specific and influenced by socio-economic development pathways. a, c, e, g** Specific electrolytic hydrogen production cost for reference, business-as-usual, 2 °C, and 1.5 °C, respectively. **b, d, f, h** Specific life cycle GHG emissions of electrolytic hydrogen production for reference, business-as-usual, 2 °C, and 1.5 °C, respectively. Geographical areas in white represent unsuitable hydrogen production locations due to spatial constraints. GHG greenhouse gas.

## Water-scarce regions

Water scarcity corresponds to an imbalance between freshwater availability and water demand, and is increasingly perceived as a global socio-environmental threat to food and energy security[22]. Since most climate mitigation solutions are water-intensive, their future adoption will commit humanity to additional water use and water scarcity[23]. Thus, water scarcity can represent a limiting factor for scaling-up hydrogen production and is examined for selected hydrogen production scenarios.

Figure 4 illustrates country-specific onshore hydrogen production via electrolysis and a global map with regions currently subject to water scarcity for the 2 °C scenario in 2050. The left subplot shows the twenty six countries with the largest hydrogen production potential, with the bar segments illustrating the amount of hydrogen produced in water-scarce regions, which are referred to as—from moderate to severe threat— moderate, significant, and severe water scarcity (based on ref. 24). Low water scarcity indicates no or negligible water scarcity. The share of hydrogen production— based on country-specific land area—in water-scarce regions is given above the overall bar segment per country.

The figure shows that large hydrogen production potentials can be found in water-scarce regions. Except for Russia, Canada, and Brazil, most other countries have large hydrogen production potentials in severe water-scarce regions, especially Australia, China, North Africa, and countries in the Middle East. Overall, more than 60% of the global electrolysis-based onshore hydrogen production potential is situated in water scarce regions. Indeed, water scarcity is expected to become an even bigger threat due to climate change and population growth[25]; hence our water scarcity analysis can be considered optimistic. Our results highlight that water consumption and scarcity should be considered in large-scale energy systems analyses to avoid a further increase in water scarcity[26], which could exacerbate food and water security[23].

## Impacts of global production

Here, we discuss the global cost and environmental implications of a future hydrogen economy under different scenarios for 2050, namely Baseline (111 Mt $H_2$ year$^{-1}$), 2 °C (451 Mt $H_2$ year$^{-1}$), 1.5 °C (364 Mt $H_2$ year$^{-1}$), and 1.5 °C – IRENA[5] (614 Mt $H_2$ year$^{-1}$). For each of these scenarios, electrolytic hydrogen demand is met by production at the best

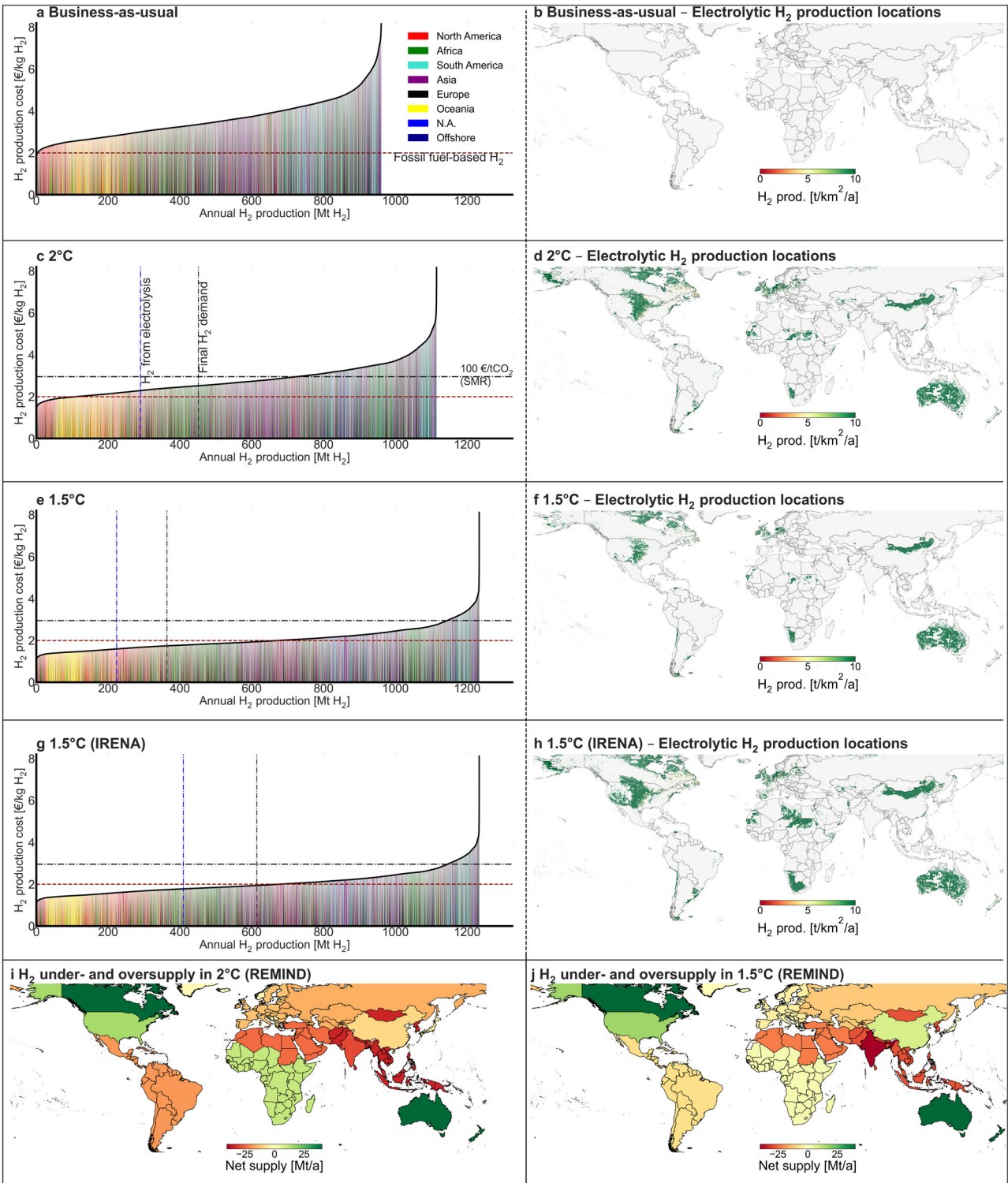

**Fig. 3 | There is a mismatch between H₂ supply and demand. a, c, e, g** Cost supply curves in year 2050 for business-as-usual, 2 °C, 1.5 °C, and 1.5 °C (IRENA), respectively. **b, d, f, h** Selected economical locations for business-as-usual, 2 °C, 1.5 °C, and 1.5 °C (IRENA), respectively. **i–j** Net H₂ supply for 2 °C and 1.5 °C, respectively. IRENA International Renewable Energy Agency, REMIND The REgional Model of INvestments and Development.

economic (=least cost) locations (i.e., spatial grid cells), and other hydrogen production routes complement electrolytic hydrogen. Overall electrolysis capacity and renewable electricity generation required for hydrogen production, costs, GHG emissions, land occupation, materials (iridium, copper, and three rare earth metals), and water consumption are quantified.

Figure 5 illustrates the results with nine stacked bar subplots, visualizing overall cost, installed capacities of technologies needed, (annualized) GHG emissions, water consumption, and the annualized utilization of copper, iridium, neodymium, praseodymium, and dysprosium. The development scenarios are presented on the *x*-axis and the impact on the latter indicators on the *y*-axis. This figure reveals

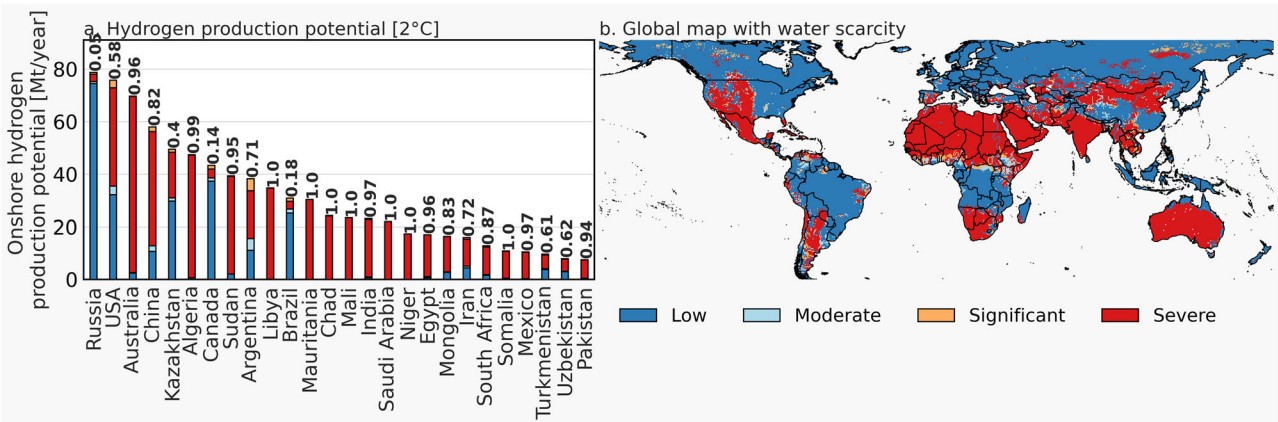

**Fig. 4 | Approximately 64% of H₂ production potentials are concentrated in water-scarce areas. a** Country specific hydrogen production potential for 2 °C scenario. **b** Global map showing water scarcity, using data from ref. 24.

significant implications for scaling-up hydrogen production in future energy systems.

## Economics and power capacities

Evaluated hydrogen (production) economies are likely trillion euro industries annually (Fig. 5). The specific hydrogen costs per unit are decreasing over time depending on the development trajectory due to associated cost assumptions and technological learning. The average specific hydrogen production cost is between 1.7–2.2 € kg⁻¹ H₂, considering all hydrogen production technologies. The installed power generation technologies slightly differ between the different pathways. However, most simulated hydrogen economies with substantial electrolytic hydrogen production require several terawatts of electrolyzers, onshore and offshore wind, and solar PV–up to 4.8 TW for electrolyzers in 1.5 °C scenarios (IRENA). In general, more onshore wind is installed as opposed to solar PV and offshore wind, mainly due to their lower levelized costs of electricity as a result of lower expected capital expenditures and higher electricity yields. The implications of installing large amounts of onshore wind (and other renewables) extend beyond economic and environmental aspects but also influence non-technical-aspects, such as the energy landscape (i.e., their scenicness), communities (local resistance), and regulatory frameworks[27].

Overall, there might be a shortage of renewable electricity in scenarios with large electrolytic hydrogen production; with renewable electricity requirements (11–20 PWh year⁻¹) higher than that of the USA (4.5 PWh year⁻¹)[28] and China in 2022 (8.8 PWh year⁻¹)[28]. The 1.5 °C scenario of REMIND deploys a substantial capacity of biomass gasification with carbon capture and storage to generate negative emissions, which leads to (average) higher hydrogen production cost compared to the IRENA scenario.

## GHG emissions

GHG emissions from future hydrogen production industries might be significant and are likely 0.7–0.9 GtCO₂-eq. year⁻¹ in ambitious climate scenarios, which inevitably require carbon dioxide removal practices to compensate for these residual emissions to reach net-zero goals[29]. Indeed, carbon dioxide removal practices are already included in the 1.5 °C scenario (REMIND) with biomass gasification with carbon capture and storage, which results in negative overall emissions of hydrogen production (around −1.1 GtCO₂-eq. year⁻¹).

Future pathways exhibit much lower specific GHG emissions (−3–1.7 kg CO₂-eq. kg⁻¹ H₂) compared to GHG emissions from hydrogen production today (15 kg CO₂-eq. kg⁻¹ H₂), which emits around 1.35 GtCO₂-eq. year⁻¹ for a much smaller hydrogen economy. It turns out that it is crucial to produce low-carbon hydrogen while minimizing H₂

leakages and reducing embodied emissions from technologies (Supplementary Note 8).

## Water and land

Water consumption is mainly driven by direct water consumption, needed for water electrolysis. However, indirect water consumption can be especially significant when relying on solar PV and biomass-based hydrogen production pathways. However, our most economical locations also select wind-powered hydrogen production, and indirect water consumption is rather small for them. In any case, roughly 13–22 billion cubic meters (bcm) of water is required for future hydrogen economies, which is comparably small (approximately 1.3–2.2%) to the annual blue water consumption of humanity (1000 bcm blue water year⁻¹)[30].

Here, one notable limitation is that we used the same biomass dataset and type of biomass, which is mainly based on sustainable forest management. Thus, the water and land impacts of biomass-based hydrogen production might be underestimated in our analysis (see the discussion). As discussed in the previous section, water consumption in water-scarce regions can still be problematic for large-scale hydrogen production deployment. Hydrogen production on offshore wind platforms or islands largely avoids freshwater utilization due to the direct desalination of seawater.

Further, we find significant land and sea area occupation requirements for electrolytic hydrogen production in ambitious climate 1.5 °C pathways, ranging from 0.15 million km² (REMIND[18,21]) up to 0.3 million km² (IRENA[5]), i.e., more than the land area of the United Kingdom. However, this figure increases significantly by 0.9 million km² if we include the land transformation over the system lifetime for biomass-based hydrogen production using carbon capture and storage in the 1.5 °C scenario of REMIND. This implies that land occupation can be a limiting factor in countries with large hydrogen production hubs or substantial biomass cultivation. Here, the maximum land utilization is limited to a maximum of 4% of the global land area (and 2% and 10% in the sensitivity analysis) to better account for limitations concerning the adoption of hydrogen production hubs, and therefore hydrogen production locations are better distributed worldwide. However, applying a higher share of land utilization most likely results in land scarcity in countries with many economical geographical hydrogen production locations.

## Materials

The integration of additional renewable energy capacity requires materials for wind turbines, batteries, solar PV panels, and electrolyzers[8,31,32]. We find that the utilization of materials might be critical for iridium and some rare earth metals, such as dysprosium.

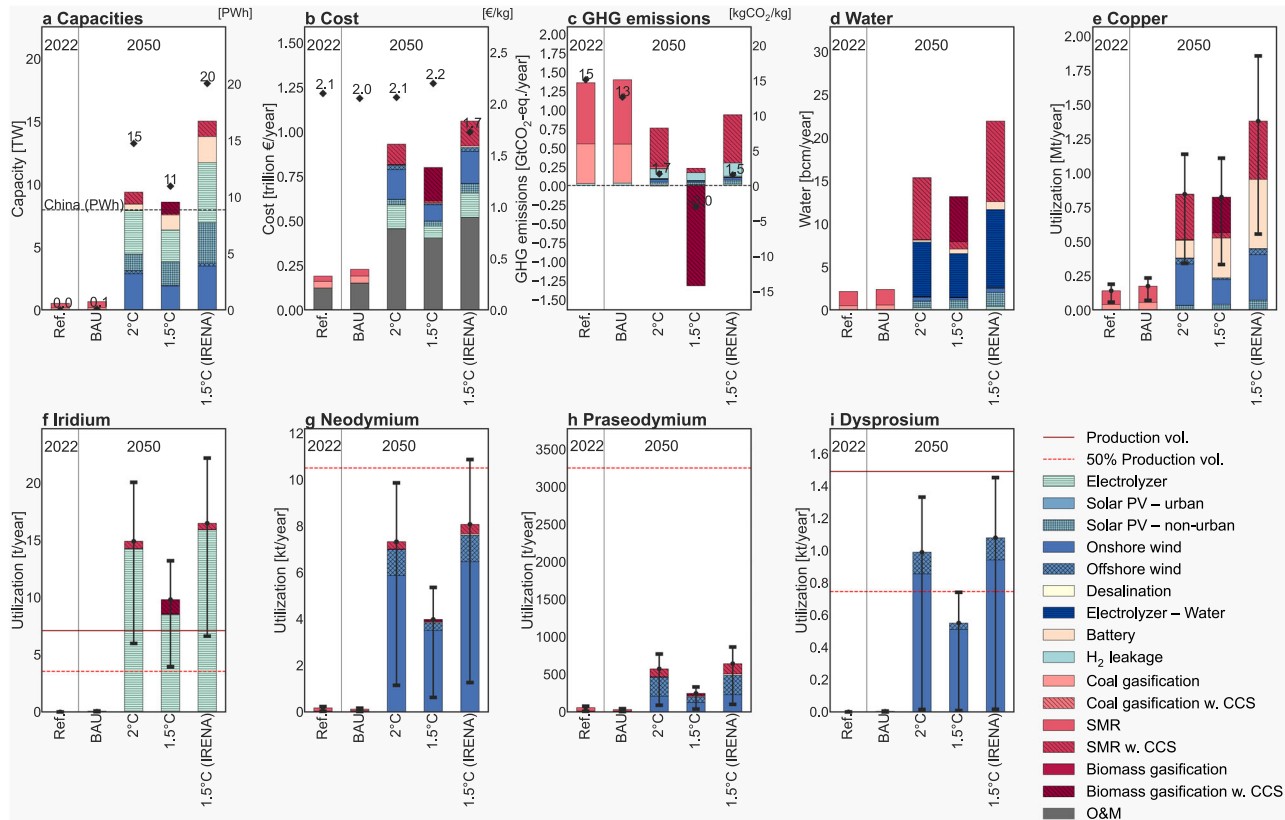

**Fig. 5 | Large-scale H₂ production might be limited by renewable electricity and natural resources. a** Installed capacities. **b** Cost. **c** GHG emissions. **d** Water consumption. **e** Copper utilization. **f** Iridium utilization. **g** Neodymium utilization. **h** Praseodymium utilization. **i** Dysprosium utilization. PV photovoltaic, CCS carbon capture and storage, SMR steam methane reforming, O&M operation & maintenance, BAU business-as-usual, Ref. Reference, IRENA International Renewable Energy Agency, GHG greenhouse gas. The error bars represent uncertainties concerning material efficiency improvements[71].

Iridium is highly utilized in the bipolar plates of PEM electrolyzers while rare earth metals are mainly utilized as permanent magnets in wind turbines today. For iridium utilization in electrolyzers, a substantial increase in material efficiency is expected and this assumption has been applied to future scenarios. Iridium production is approximately 7.1 tonnes year⁻¹ [33] nowadays, which could lead to a shortage due to future iridium utilization in electrolyzers of 10–16 tonnes year⁻¹ in ambitious climate scenarios. In addition, the utilization of rare earth metals might be a limiting factor for future deployment of wind-based hydrogen production, although our scenarios demonstrate that improved material efficiency could substantially reduce metal criticality (see the error bars in Fig. 5). For example, the annual production volume of dysprosium is approximately 1490 tonnes year⁻¹, while we find a future dysprosium requirement of up to 1080 tonnes year⁻¹. Thus, this material might be considered critical—reaching a demand near to current dysprosium supply—when there is a lack of material efficiency in future hydrogen economies, and when mainly powered by wind energy sources. Note that the future global demand for some rare earth metals analyzed, e.g., neodymium and dysprosium, are expected to relatively increase up to more than tenfold by 2050 due to unprecedented demand by competing energy sectors[32], which might lead to challenges in upscaling production capacities at the required pace and to supply chain disruptions in the worst case scenario.

Although not considered in the main analysis, lithium metal might emerge as another critical material in scenarios installing extensive battery electricity storage. To illustrate this, a total battery capacity of approximately 2 TWh is deployed in the 1.5 °C (IRENA) scenario for hydrogen production configurations only. Assuming an optimistic future utilization of 0.15 kg/kWh for nickel manganese cobalt batteries[34], this would require approximately 300,000 tonnes of lithium. Currently, lithium production is around 100,000–130,000 tonnes year⁻¹ [35,36]. It is, however, unlikely that the total extraction of materials occurs in a single year. Further, material criticality might be reduced by improved material efficiency, recycling of metals and materials, annual production volumes can be increased, and alternative technologies or material substitutes might be developed[37]. These factors could collectively decrease the potential criticality of lithium and other materials by 2050.

In summary, the main factors driving our results are the future global macroeconomic development (e.g. influenced by technology development and learning), hydrogen demand (e.g. influenced by policy and collaboration), and potential resource constraints, such as material and land availability. For our global cost analysis of electrolytic hydrogen production, the capacity factor of hydrogen production (and renewables) has the highest impact (Supplementary Fig. 36). The investments in electrolyzers and onshore wind, as well as the weighted average cost of capital (WACC) for these technologies, have a moderate impact. The other parameter changes have a low impact on cost. For our global analysis of GHG emissions from electrolytic hydrogen production, the capacity factor of hydrogen production (and renewables) has the highest impact (Supplementary Fig. 36). Hydrogen leakage and embodied emissions of solar PV and onshore wind, as well as their component lifetimes, have a moderate impact. The other parameter changes have a low impact on GHG emissions. Hydrogen leakage has a substantial effect due to the potential (indirect) GHG impact of leaking hydrogen into the atmosphere. Embodied emissions from solar PV and onshore wind are attributed to the manufacturing processes of PV wafers and wind turbines, respectively.

## Discussion

This study shows that a zero-emission hydrogen economy does not exist and likely exhibits around 1 $GtCO_2eq.$ $year^{-1}$, even in the most ambitious decarbonization scenarios. One exception is the 1.5 °C pathway of REMIND, resulting in negative emissions due to the significant integration of biomass-based coupled hydrogen production with carbon capture and storage. In fact, hydrogen production from biomass coupled with CCS is the only production pathway that can generate net negative emissions[14,16] within this work. These findings have some important implications. First, it is better to refrain from the green hydrogen concept and instead apply an emission factor and/or a metric that considers overall environmental burdens to improve the comparison of low-carbon hydrogen production pathways, which is in line with a recent report by the International Energy Agency (IEA)[38]. Second, applying a single emission factor for renewables per unit of energy used for electrolytic hydrogen production must be prevented when adopting a global scope, as this is inappropriate for capturing location-specific energy yields of renewables and thus environmental burdens associated with such hydrogen.

We show that the large-scale global deployment of electrolytic-based hydrogen production likely exhibits environmental trade-offs due to substantial expansions required for renewable energy technologies. This expansion requires significant natural resources in terms of water, land, and critical materials, although overall mining of materials in a low-carbon economy is likely significantly smaller compared to current fossil fuel-based economies[39]. Competition for renewable electricity and natural resources becomes a critical aspect in net-zero emission pathways due to increased electricity, land, water, and material demand from competing electricity-intensive industries, such as battery electric vehicles and some carbon dioxide removal options, e.g., direct air capture[40,41]. Future efforts should aim to integrate such energy sectors into geospatial analyses to provide more reliable estimations of the overall impacts of materials and electricity demand on a global level.

Our hydrogen scenarios, derived from integrated assessment model (IAM) pathways and IRENA, are inherently uncertain due to potential technology breakthroughs and policy development until 2050. Here, we focused on low-carbon hydrogen production mainly via PEM water electrolysis using wind and solar PV as power sources. However, due to the development of alternative electrolyzers and energy sources as well as policy, current low-carbon hydrogen production energy systems might be replaced by more cost-efficient pathways, although they are likely to exhibit other environmental trade-offs. Nevertheless, our analysis reveals that novel hydrogen production pathways should be assessed using global scope to identify potential environmental trade-offs and limitations.

We further find region-specific limitations for the large-scale deployment of hydrogen production. Foremost, land and water requirements might be limiting factors for the deployment of hydrogen production on a regional scale, affecting more than 60% of the suitable hydrogen production regions due to water scarcity.

Hydrogen production from the air might be a promising technology to overcome such water scarcity, although technological readiness is currently low[42]. While the overall additional water demand from future hydrogen economies is generally modest (13–22 bcm water $year^{-1}$) in comparison with other human water uses, it could still be a bottleneck for expanding hydrogen production in regions subject to water scarcity or reliant on biomass cultivation[22,43]. In our analysis, land and sea area occupation requirements for electrolytic hydrogen production are in the range of 0.15–0.30 million $km^2$ over the system lifetime—max. surface corresponding to the land area of Poland—in ambitious climate pathways. This figure increases significantly by almost one million $km^2$ if we include the land transformation needed for biomass-based hydrogen production in a 1.5 °C scenario of REMIND. Thus, land occupation could be another limiting factor for

countries with large-scale hydrogen production ambitions via biomass sources.

One limitation of this study is that we use single life cycle inventories for other hydrogen pathways beyond water electrolysis, due to limited data availability of location-specific data and inventories. A critical reflection on potential implications is given in the Supplementary Note. For example, some recent studies report much higher water footprints of biomass gasification, up to more than 3400 kilogram $H_2O$ per kilogram $H_2$[44], due to the exclusion of green water (soil moisture from rainfall[45]) consumption in this study. Applying such a specific water consumption results in a substantial increase of global water consumption in the 1.5 °C scenario up to 413 bcm for biomass gasification with carbon capture and storage only. Similarly, a generic hydrogen leakage factor of 2.5% is used in the main analysis for all hydrogen production pathways, applying a higher hydrogen leakage factor results in substantial additional radiative forcing[46]. Similarly, no economic value has been given to oxygen as a by-product of hydrogen production since we focus on off-grid hydrogen production systems. A large-scale demand and supply chain of oxygen is not developed in most geographical regions (except in cities near hospitals), especially in off-grid regions. However, oxygen can have a substantial economic value between a large range 0.04[47]–3[48] euro/kg $H_2$, where the upper level represents oxygen utilization in hospitals. Thus, considering oxygen as a useful by-product could result in substantial revenues for hydrogen production. And it could also reduce its LCA-based greenhouse gas emissions, especially when applying economic allocation given that 8 kg of $O_2$ is produced per kilogram of hydrogen production.

Reported water consumption and land utilization figures align with recent analyses, although these only focus on electrolytic hydrogen production and are therefore difficult to compare with our analysis. Blanco[49] reports a water consumption of 25 bcm for an electrolytic hydrogen economy of 617 Mt $H_2$ $year^{-1}$. A large range of 3.2–95.6 bcm $year^{-1}$ of water withdrawals is reported in Tonelli et al.[15], highly depending on the hydrogen production demand scenario and solar PV or onshore wind energy used for hydrogen production. In addition, ref. 15 estimated a land utilization of 0.09–0.6 million $km^2$ for solar PV panels and 1.9–13.5 million $km^2$ for onshore wind turbines for electrolytic hydrogen demands of 92–646 Mt $year^{-1}$. This is slightly higher than the land transformation found in our analysis, likely due to (i) smaller electrolytic hydrogen production quotas in our analysis, (ii) the use of economical hybrid hydrogen production systems and corresponding location-specific environmental impact factors, highly influenced by location-specific renewable energy yields. Further, we include other hydrogen production routes, such as steam methane reforming, coal gasification, and biomass gasification with carbon capture and storage, which makes direct comparisons challenging. Alternative hydrogen production pathways are available to produce low-carbon hydrogen, such as pyrolysis and using other low-carbon electricity sources (e.g., nuclear or hydropower) for water electrolysis, which are not included in this study.

Compared to previous studies, we reveal two additional important findings: (i) land requirements and water consumption are significantly larger when integrating biomass-based hydrogen sources (especially when including green water), and (ii) regional water scarcity could be potentially mitigated by offshore hydrogen production and/or the utilization of water desalination. Water desalination could be a promising technology to purify seawater at locations with access to brackish water, and is associated with marginal GHG emissions[8] and a cost of less than 0.02 € $kg^{-1}$ $H_2$[8,50], which typically represents less than 1% of hydrogen production cost[8,50]. However, appropriate brine disposal in marine environments remains critical, as it can still impose significant environmental burdens[50,51]. Additionally, reusing and cleaning wastewater could be another option to reduce water scarcity, although PEM electrolyzers require very high water purity. Deploying

large-scale offshore hydrogen production platforms might be a promising way to reduce potential onshore land and water scarcity from large-scale hydrogen production. One notable limitation of this study, and previous studies, is that indirect water flows (and land occupation) are not attributed to the location of the indirect impact. More sophisticated modeling efforts are critical to trace such flows, given that water and land use-related impacts are very location-specific.

We find evidence for a mismatch between future economical hydrogen supply and demand. Thus, facilitating trade of hydrogen-based products—i.e., ammonia[52,53], methanol[54], and synfuel[55]—will be key to future low-carbon energy systems. This requires the development of dedicated distribution and transport networks of such hydrogen-based products[54]. While our focus is on hydrogen production, decision-makers should develop effective policy measures to stimulate distribution networks of hydrogen-based products and might need to re-evaluate the distribution of future hydrogen-intensive industries. As this study excludes further hydrogen compression, transportation, or end-uses of hydrogen, including these aspects will likely further increase cost and environmental burdens[9,11]. One approach would be to couple our cost-effective hydrogen production locations with frameworks that optimize hydrogen transportation supply chains. This might lead to a more complete view of the overall life cycle impacts of hydrogen and allow for identifying supply chain improvements and associated cost reductions.

Overall, our findings highlight that a future hydrogen production economy, largely based on water electrolysis and biomass pathways, exhibits much lower GHG emissions than fossil-fuel-based hydrogen production. However, they should be quantified using a life cycle perspective to explore economic and environmental impacts and their interactions. Future global assessments should consider such environmental trade-offs and not only focus on cost and (operational) GHG emissions. In fact, environmental trade-offs should be carefully analyzed when scaling up hydrogen production to avoid undesirable geopolitical and supply chain consequences concerning materials, water, land, and renewable energy sources.

## Methods

We consider electrolytic hydrogen production, coal gasification, biomass gasification, and steam methane reforming with and without carbon capture and storage as hydrogen production pathways. The future hydrogen production mixes— obtained from REMIND[18,21] and the scenario of IRENA[5]—are complemented by electrolytic hydrogen production until hydrogen demand is met. Thus, our main focus is on optimal electrolytic hydrogen production to complement these future hydrogen production mixes.

In short, this study uses a geospatial analysis to quantify; (i) electrolytic hydrogen production based on the potential of renewables, (ii) the cost of hydrogen production, and (iii) environmental burdens of hydrogen production considering GHG emissions, water consumption (under water scarcity), potentially critical materials, and land occupation. Here, the methodologies and data requirements are discussed. We focus on a reference situation (2022) and on four different prospective scenarios to determine the implications of a potential large-scale hydrogen economy in 2050.

### Evaluation framework and scope

Future climate mitigation scenarios (Fig. 1g) illustrate that electrolytic hydrogen production is the most dominant future low-carbon hydrogen production pathway in ambitious climate scenarios[5,18]. Further, some other potentially prominent low-carbon hydrogen production pathways exhibit ambiguous climate change impacts, especially blue hydrogen[56,57]. Blue hydrogen is hydrogen production via steam methane reforming combined with CCS, but the climate impacts are uncertain due to potential fugitive methane emissions, the capture rate of CCS, and the global warming potential metric applied. Thus,

blue hydrogen is often considered as a 'bridging technology' filling a gap in low-carbon hydrogen demand as long as electrolysis is not sufficiently scaled up[56,58]. Our geospatial analysis therefore focuses on low-carbon electrolytic hydrogen production—considering solar PV, onshore wind, and offshore wind—by performing a detailed pixel-based geospatial analysis at $0.25° \times 0.25°$ resolution that assesses potentials and location-specific costs and environmental burdens. This identifies the most suited hydrogen production locations taking into account local boundary conditions and constraints, such as renewable energy yields and water availability. If there is significant solar PV and onshore wind potential, onshore hybrid hydrogen production configurations are designed based on non-linear trend correlations using energy system optimization (Section on 'Hybrid hydrogen production systems') that considers location specific renewable energy potentials from wind and solar PV. Electrolytic hydrogen production is complemented by other hydrogen production pathways in future hydrogen production scenarios, such as hydrogen from biomass gasification and steam methane reforming. All global locations are theoretically considered for electrolytic hydrogen production (with solar PV and wind) but might be excluded due to spatial constraints, such as the type of land, altitude, and terrain slope (Section on 'Potential of renewables and land utilization').

Figure 1 gives a graphical overview of the methodology. The upper part of Fig. 1 illustrates the system boundaries of two configurations: onshore and offshore. Onshore hydrogen production can utilize solar PV and onshore wind electricity systems in the electrolyzer and store electricity in a battery. None of these configurations are connected to the power grid as, considering our global scope, power grids are often not present. Electricity from the grid is also often associated with substantial GHG emissions, at least today and likely within the next decade(s). Local deionized tap water is considered for the water requirements of the electrolyzer. Offshore hydrogen production is solely based on offshore wind electricity production. Here, water must be desalinated with a water desalination plant[59].

### System boundaries and life cycle analysis

All supply chain activities are included from the point of producing energy technologies (i.e., including embodied emissions) up to the point of hydrogen production of 25–30 bar. Further hydrogen compression, transportation, distribution, and use are not included in this analysis, thus, it refers to a cradle-to-gate life cycle analysis (LCA). Environmental burdens are quantified for each activity in the supply chain and are considered in the geospatial analysis. The life cycle inventory (LCI) of these activities is given in Supplementary Table 1, and a contribution analysis on all selected environmental impact categories is provided in Supplementary Note 7. For our LCA, we use ecoinvent 3.9.1[60,61] (system model: "allocation, cut-off by classification") as the background LCA database using the open-source Python package brightway2 to calculate LCA results[62]. The open-source Python package premise (v.2.0.0)[19] is used to import the up-to-date LCI of emerging technologies and to modify the ecoinvent background LCA database using scenarios from the integrated assessment model REMIND to perform prospective LCA for the year 2050 scenario analysis[18,21].

Electrolytic hydrogen production is assumed to be produced with PEM electrolyzers, to complement the other hydrogen production pathways in Table 1. PEM electrolyzers are commercially available and have a fast response to renewable electricity generation[8,11]. Further, PEM electrolyzers are likely the best economic and environmental hydrogen production option in the long term, compared to alkaline and solid oxide electrolyzers[63,64]. The typical output pressure from PEM electrolyzers is assumed to be in the range of 25–30 bar[11]. The water-splitting process requires a significant amount of electricity (48–56 kWh kg$^{-1}$ H$_2$), which can be supplied by solar PV, onshore wind, and offshore wind electricity sources for offshore locations. A life cycle

analysis approach is applied to determine the costs and environmental burdens of cradle-to-gate hydrogen production, which means that all upstream activities are included up to compressed hydrogen of 25–30 bar. We aim to quantify if supply chain emissions—especially from the construction of energy technologies (i.e. embodied emissions)—can reduce the environmental merits of electrolytic-based hydrogen with solar and wind power.

Environmental impact categories are adopted from the developer environmental footprint method (EF, v3.1)[65], except for climate change impacts where we include the IPCC 2021 global warming potential (GWP100a, including hydrogen and biogenic $CO_2$) as the impact category using 'premise gwp'[19]. The climate change impact category includes characterization factors for biogenic $CO_2$ and hydrogen (GWP100 factor of 11), which we add to the original IPCC characterization factors to be able to represent carbon dioxide removal from the atmosphere for wood gasification with CCS and to take into account the latest evidence regarding radiative forcing of hydrogen emissions. Net water consumption is estimated using the corresponding midpoint impact category from ReCiPe 2016 (v1.1 (20180117))[66,67]. Both direct and indirect water consumption are considered, where direct water refers to water consumption that takes place onsite during hydrogen production at the hydrogen production facility: for example, for the water splitting process (in water electrolysis). On the contrary, indirect water footprints refer to all other water consumption that takes place at other locations throughout the upstream hydrogen production supply chain. Green water consumption is not included in this impact.

Here, special attention is given to water consumption and scarcity of a future hydrogen economy. A prospective hydrogen economy commits to additional water consumption, that can potentially increase water competition between natural ecosystems and other human uses. The large-scale deployment of hydrogen production in non-coastal regions in combination with expected additional water demand due to climate change, electrification, population growth, economic development as well as agricultural intensification might result in water scarcity[22,68,69]. The twin challenges of mitigating climate change and water scarcity are therefore competing factors in reaching net zero emissions systems. Therefore, we obtain a geographical map of water scarcity from ref. 24 to determine the share of hydrogen production subject to water scarcity. Indeed, the map represents the current situation, and this situation will likely change in the coming decades due to further global changes concerning population, land use changes, and climates, which will likely increase global water scarcity. Thus, our water scarcity analysis can be considered optimistic.

Further attention is given to material demand for a set of potentially critical materials. For material demands of key technologies, we obtain averaged material demand from literature (ref. 70) concerning copper[32], iridium[8], and three rare-earth metals: neodymium, dysprosium, and praseodymium[32,71]. Copper and these rare earth metals are selected as they might represent bottlenecks for large-scale hydrogen production, especially energy originating from wind energy sources[71–73].

## Hybrid hydrogen production systems

In this study, solar PV and onshore wind can be installed in onshore land regions to supply electricity for water splitting, which induces the complexity of the potential implementation of hybrid energy systems that can install both wind and solar PV capacity. Such energy system configurations can be installed as cost-optimal solutions to use available renewable energy sources optimally[8]. Thus, this requires (i) the determination of optimal wind to solar PV shares, based on wind and solar PV energy availability in a spatial grid cell, and (ii) the corresponding battery capacity used to store intermittent generation. To solve this problem, we use energy system optimization for a hundred global locations that exhibit high renewable energy potentials for onshore wind and/or solar PV. A mixed integer linear program (MILP) is developed to design hybrid hydrogen production system configurations based on annual costs considering onshore wind, solar PV, and lithium-ion batteries. The complete procedure is explained in Supplementary Note 2. The outputs of the hundred optimized case studies result in non-linear trend line correlations, which are used in the geospatial analysis to identify and estimate (near-)optimal shares of wind vs. solar PV, electrolyzer capacity, and battery capacity in all spatial grid cells.

## Potential of renewables and land utilization

Hydrogen production via water electrolysis requires a substantial amount of electricity, which must be covered by low-carbon energy sources, such as renewables, to produce very low-carbon hydrogen compliant with a net-zero global economy[8,12]. However, the installation of renewables is typically constrained by several factors, particularly land occupation and social acceptance. Geospatial analysis, with pixel-based details, is applied to determine the amount of land available for renewable electricity production, which is required for hydrogen production via water electrolysis.

To achieve this, land use factors are applied, using land use types from the Copernicus global land use coverage database[74]. These global land use types are coupled to a suitability factor for renewable energy generation for solar PV and onshore wind. This suitability factor is included in parameter $\alpha_{land}$ of Eq. (1), the latter equation is used to determine these renewable energy potentials. The suitability factors are obtained (based on refs. 15,75) for each suitable land type to determine the wind potential and solar PV potential, which can be found in Supplementary Table 3. To consider competing purposes of renewable energy generation and factors such as social acceptance, we multiply the latter utilization factors with 4% (included in parameter $\alpha_{land}$) to obtain more realistic renewable energy potentials for hydrogen production. Indeed, this is a subjective choice; however, it has been chosen in a conservative way to consider potential social acceptance issues and other land area constraints. Similar land utilization percentages are used in previous geospatial analyses (e.g., in ref. 15). Further, additional sensitivity analyses with different land utilization factors (for 2% and 10%) are presented in Supplementary Note 8.

In addition, binary land masks are applied to prevent the installation of renewables and hydrogen production in unsuitable regions,

## Table 1 | Hydrogen production scenarios considered

| Scenario | Year | Mt $H_2$ year$^{-1}$ | Main technologies | Ref. |
|---|---|---|---|---|
| Reference | 2022 | 90 | Coal gasification and steam methane reforming | 88 |
| Business-as-usual | 2050 | 111 | Coal gasification and steam methane reforming | SSP2-Base[18,21] |
| 2 °C | 2050 | 451 | Steam methane reforming with carbon capture and storage and water electrolysis. | SSP2-PkBudg1150[18,21] |
| 1.5 °C | 2050 | 364 | Biomass gasification with carbon capture and storage and water electrolysis. | SSP2-PkBudg500[18,21] |
| 1.5 °C (IRENA) | 2050 | 614 | Steam methane reforming with carbon capture and storage and water electrolysis. | IRENA[5] |

*IRENA* International Renewable Energy Agency, *SSP* shared socioeconomic pathway.

such as protected regions (based on data of ref. 76), altitudes higher than 2000 m (due to infrastructure constraints, based on data of ref. 77), non-exclusive economic or unclaimed zones (based on data of ref. 78), low capacity factors of renewables (solar PV < 0.05 and onshore wind <0.20), steep terrains (slopes >15°,[79] based on data of ref. 80), offshore regions with sea depths more than 50 meters that are assumed unsuitable for offshore wind (based on data of ref. 77), and with low capacity factors of offshore wind (offshore wind <0.25). To do so, a binary parameter y is used in Eq. (1), which is '1' when the area is suitable for renewable energy generation and '0' otherwise. The global shapefile of the World Bank is used for country boundaries[81,82]. Further, we assume that onshore wind and ground-mounted solar PV cannot be installed in urban regions, solely residential solar PV can be installed in urban regions.

Renewable energy generation is location-specific and depends on the local boundary conditions in terms of for example solar irradiation and wind speed. Therefore, location-specific capacity factors for onshore wind, offshore wind, and solar PV are obtained from the SolarAtlas[77] and WindAtlas[83,84]. It is worth noting that solar PV capacity factors are only covered at -60–60 degrees latitude due to a lack of data from the SolarAtlas and other sources. Thus, the potential of electrolytic hydrogen production using solar PV electricity is limited to these latitudes. Class "IEC1" is considered for onshore wind as this represents wind turbines with a lower rotor diameter compared to other classes, such as "IEC3" (increased wind power output) which has been used for offshore wind capacity factors[84]. Offshore wind potentials are covered up to 200 km from the shoreline[83].

Overall, the individual renewable energy potentials $E$ (in MWh yr$^{-1}$) of energy technologies are calculated and shown as in Eqs. (1) and (2), for each spatial grid cell (0.25° × 0.25°, surface area covered in parameter $A_{grid}$):

$$s_i = y\alpha_{land}A_{grid}\rho_i, \qquad (1)$$

$$E = 8760(1 - \beta_i)\varepsilon_i s_i. \qquad (2)$$

where $s_i$ is the amount of renewable energy capacity installed in a specific grid cell of technology $i$ [MW], $\rho_i$ is the specific power density of technology $i$ [MW/km²], $\varepsilon$ is the capacity factor of technology $i$ in a grid cell [−], $\beta_i$ is the curtailment ratio of technology $i$, and 8760 are the annual number of hours [h year$^{-1}$]. The specific power densities for solar PV, onshore wind, and offshore wind can be found in Supplementary Table 2.

The parameter land utilization, denoted by $\alpha_{land}$, includes consideration of the land suitability factor (depending on land use type), the share of land available for electrolytic hydrogen production in a grid cell (4% assumed in the main analysis, and 2% and 10% in a sensitivity analysis), and the share of solar PV vs. onshore wind that can be installed, which is based on optimization of hybrid energy systems (Section on energy system optimization). The annual amount of renewable energy generation is used to determine the amount of possible hydrogen production in a grid cell considering the energy density of hydrogen (120 MJ kg$^{-1}$ H$_2$) and the efficiency of the PEM electrolyzer.

### Techno-economic assumptions and energy scenarios
Techno-economic data is presented in Supplementary Table 2. The country-specific WACC for solar PV, onshore, and offshore wind is obtained from ref. 85 Average WACC are considered for countries without WACC data.

Here, we consider a broad space of potential development trajectories. Therefore, various shared socioeconomic pathway 2 (SSP2) scenarios—based on the outputs of IAMs—are chosen since they represent middle-of-the-road pathways to socio-economic

(intermediate) challenges for adaptation and mitigation[86,87]. As such, SSP2 represents a pathway characterized by moderate challenges and uncertainties, which ensures a balanced scenario between potential extreme future development trajectories. The following scenarios are included:

First, the Reference (2022) scenario represents the current global economy. For this purpose, the current costs of technologies are applied and the ecoinvent 3.9.1 database is used as background LCI. For the current situation, an overall production of 90 Mt H$_2$ year$^{-1}$ [88] has been considered.

Second, the Business-as-usual (BAU) scenario represents a possible future global economy in 2050 using the current socio-economic trend, i.e., a business-as-usual pathway that extrapolates historical developments. For this purpose, the background LCA database is modified using the *SSP2-Base* scenario from REMIND[18,21]. This pathway of REMIND corresponds to an overall production of 111 Mt H$_2$ year$^{-1}$.

Third, the 2 °C scenario represents a future global economy in 2050 that limits global warming to 2 °C. For this purpose, the background LCA database is modified using the *SSP2-PkBudg1150* scenario from REMIND, which corresponds to an overall production of 451 Mt H$_2$ year$^{-1}$ [18,21].

Fourth, the 1.5 °C scenario represents a future global economy in 2050 that limits global warming to 1.5 °C. For this purpose, the background LCA database is modified using the *SSP2-PkBudg500* scenario from REMIND, which corresponds to an overall production of 364 Mt H$_2$ year$^{-1}$ [18,21].

Fifth, an alternative 1.5 °C (IRENA) scenario represents a future global economy in 2050 that limits global warming to 1.5 °C. For this purpose, the background LCA database is modified using *SSP2-PkBudg500* scenario from REMIND, but corresponding to an overall production of 614 Mt H$_2$ year$^{-1}$ [5]. The 1.5 °C (IRENA) pathway has substantial hydrogen integration due to the assumption that hydrogen will be produced in least-cost locations and transported—likely converted to liquid hydrogen, ammonia, or synthetic fuels—to demand centres.

The chosen future scenarios differ substantially regarding overall hydrogen requirements in 2050. This requirement is, among other factors, mainly driven by climate goals, assumed technology developments and degree of direct electrification of energy end-use application. Here, 1.5 °C (IRENA) likely represents the "upper level" of future hydrogen requirement, while the BAU scenario exhibits the "lower level" of future hydrogen requirement. Reality will be likely in between these pathways, potentially represented by the other 1.5 °C and 2 °C scenarios.

The size of these hydrogen economies and shares of different hydrogen production routes considered are illustrated in Fig. 1. Hydrogen requirements are obtained from REMIND[18,21], considering the shares of hydrogen from continents in terms of final energy. The total hydrogen demand is, however, taken from hydrogen as a second energy carrier since hydrogen is also used in hydrogen-based fuels, such as synfuels. For simplicity, in this study, hydrogen supply equals demand. Future work should consider more sophisticated hydrogen demand scenarios.

### Main performance indicators
The main economic performance indicator used is the levelized cost of hydrogen production ($C_{H2}$) per grid cell, which is calculated by considering annualized investments ($C_{inv,an}$), operation & maintenance costs ($C_{op}$), and fixed operation & maintenance costs ($C_{om}$), divided by the annual hydrogen production rate in a grid cell ($H_{2,total}$)[8].

$$C_{inv,an} = \sum_{i=1}^{M} \frac{\gamma(1+\gamma)^L}{(1+\gamma)^L - 1} C_{inv}^i, \qquad (3)$$

$$C_{H2} = \frac{C_{op} + C_{inv,an} + C_{om}}{H_{2,total}}. \qquad (4)$$

where $\gamma$ is the WACC [–], the set of technologies (M) in the hydrogen production system is indexed with $i \in \{1, 2, \ldots, M\}$, and $L$ is the lifetime of the hydrogen production system. The lifetime of the hydrogen production system is set to 30 years.

Equations (5) and (6) express the environmental impacts of the operation ($G_{op}$, such as hydrogen leakages) and embodied emissions ($G^i$) of energy technologies, and the overall environmental impact of hydrogen production per kilogram of hydrogen produced per grid cell ($G_{H2}$), respectively.

It is important to highlight that $G_{inst}$ encompass the impact of an environmental impact category, and can thus be calculated separately for GHG emissions, land occupation, and the materials needed for manufacturing and replacements of an energy technology. Thus, for each energy technology, the LCA impacts are calculated and multiplied by their installed capacities considering their lifetime[89].

$$G_{inst} = \frac{\sum_{i=1}^{M} G^i \frac{L}{L^i}}{L}, \qquad (5)$$

$$G_{H2} = \frac{G_{ins} + G_{op}}{H_{2,total}}. \qquad (6)$$

where $L^i$ is the lifetime of technology $i$ [year].

The life cycle inventory used (from background LCA database ecoinvent 3.9.1[61] using system model "cut-off by classification") to calculate LCA impacts is presented in supplementary Table 1.

Finally, a graphical overview of the methods is presented in Fig. 1 of the manuscript. Further details regarding assumptions and methods are given in the Supplementary Notes.

## Data availability
The data on electrolytic hydrogen production cost and GHG emission data (Fig. 2) have been deposited, for all scenarios, in a Zenodo repository: 10.5281/zenodo.10244447. Further data from this study can be found in the supplementary information and the references. Any additional data supporting this study's findings are available from the corresponding author (T.T.) upon request.

## Code availability
The script to generate the main figures is provided in a Zenodo repository: 10.5281/zenodo.10244447. Additional scripts supporting this study's findings are available from the corresponding author (T.T.) upon request.

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

## Acknowledgements

Financial support has been provided by the PSI ESI platform and the project SHELTERED (C.B.), funded by the Swiss Federal Office of Energy (SFOE). The authors thank Alvaro Jose Hahn Menacho (PSI) for advice and data concerning the material utilization of renewable energy technologies. Further, the authors thank the support of Alejandro Christlieb Picazo (ETH Zurich) with data collection and visualization.

## Author contributions

T.T. Conceptualization, Software, Formal analysis, Methodology, Visualization, Writing—original draft. L.R. Conceptualization, Methodology, Writing—original draft. C.B. Conceptualization, Writing—review & editing. R.M. Writing—review & editing.

## Funding

## Competing interests

The authors declare no competing interests.
