## [Peer Review File · Nature Communications]

REVIEWER COMMENTS

Reviewer #1 (Remarks to the Author):

The authors address some of the most pressing questions regarding the potential and implications of hydrogen as an energy carrier in the context of decarbonization. They highlight in particular: the challenge of water and land scarcity, life-cycle emissions of hydrogen production and issues of material scarcity. The authors also touch on other issues, such as the risks of hydrogen leakage. While these concerns are widely discussed in the literature, there is still a scarcity of rigorous, quantitative studies on these issues and what they might imply. In this vein, this paper is a welcome contribution, as it sets out to provide such a quantitative assessment. However, in my view, the paper is ultimately over-ambitious in trying to provide a meaningful quantitative contribution to all the different dimensions listed above. As a result, the paper does not address a lot of important questions regarding assumptions, data, sensitivities, etc. in detail. For instance, it is not clear how the authors arrive at the substantial GHG emissions from renewable hydrogen production. There is a reference to Life-cycle inventories. However, there is no discussion of these inventories and how they may change in the context of decarbonization. Similarly, impacts on water are addressed, but implications and options - such as desalinization - are not addressed in detail. As a result, the paper does not provide substantial new insights into the challenges of scaling-up hydrogen production, but rather restates issues that are well-known in the debate. In my view, the paper could make a much more substantial contribution if the authors would choose to focus on one of the issues in more detail and provide a more nuanced discussion of the specific issue, including the specific design of the assessment, data uses, sensitivities and uncertainties in that area (ideally without having to go to the supplemental material). This way readers could obtain detailed insights into the state of current knowledge in the chosen field and the authors could provide a nuanced discussion of implications. Given the various different topics the authors attempt to address, this is not possible. Hence, the value of the paper is substantially lower.

Reviewer #2 (Remarks to the Author):

The authors provided an extensive economic-environmental assessment of PV- and wind-based green hydrogen production considering multiple scenarios of demand in the year 2050. The manuscript is very well written, and the results are thoughtfully analyzed. This study should be valuable to many stakeholders and decision-makers on different levels. I have no significant comments on its content. Hence, I believe it can be published as is.

Reviewer #3 (Remarks to the Author):

Manuscript Title: Future hydrogen economies imply environmental trade-offs and a supply-demand mismatch

The authors carried out an in-depth survey on the critical aspects of future hydrogen economies, emphasizing the need to replace current fossil fuel-based hydrogen production with low-carbon alternatives for effective decarbonization. The study quantifies the costs and environmental impacts associated with large-scale hydrogen economies, projecting demand scenarios for 2050. The paper emphasizes the necessity of a comprehensive understanding, considering various production technologies, future scenarios, and both cost and environmental factors. I recommend this article with minor revisions.

The manuscript is written clearly; however, the following points should be clarified or added.

1. Non-linear trend correlations have critical limitations, including the impact of outliers, context specificity, hidden variables, and attractor reconstruction in time series data. These limitations need to be justified. Additionally, the manuscript should provide a more comprehensive explanation of the "novel non-linear trend correlations" method, highlighting its advantages over alternative approaches.
2. The scenarios considered for global scale-up of hydrogen production are well-defined. However, a more extensive discussion on the rationale behind choosing these specific scenarios, including the underlying assumptions and potential variations, would add depth to the research.
3. It is important for the authors to discuss the uncertainty in the impacts of climate change, especially regarding blue hydrogen. Including a more detailed discussion on how various other hydrogen production methods might affect climate change would offer a more comprehensive perspective, providing strong justification for choosing green hydrogen for a global study.
4. The author mentioned environmental trade-offs, but a more elaborate discussion, especially regarding the considerable potential of wind energy, is crucial for understanding the true environmental footprint of each method.
5. The authors should discuss the potential impact of future trends, such as advancements in technology and changes in policy, on the presented results.
6. More detailed information about the data sources and modelling assumptions used in the analysis should be included.
7. Explain the specific methods used for cost and GHG emission calculations in different regions.
8. Discuss potential limitations of the analysis, such as uncertainties in future scenarios and data availability.
9. Discuss the potential for green hydrogen trade between regions with abundant resources and those with high demand.
10. Explore technological advancements that could mitigate water scarcity issues in hydrogen production (e.g., seawater desalination, wastewater reuse).

11. Including land transformation for biomass production significantly increases land use requirements. This should be highlighted as a potential limiting factor for countries with large hydrogen ambitions.
12. The concept of limiting land utilization to 2% seems arbitrary. Justifying this limit based on specific land cover data would strengthen the argument.
13. The criticality of specific materials, particularly iridium and dysprosium, is well-discussed. Expanding on potential mitigation strategies, such as material substitution, recycling, or improved efficiency, would be helpful.
14. The paper could benefit from a more thorough comparison with relevant existing studies, highlighting the unique contributions and potential areas for further research.
15. The authors should refine land use factors and curve-fit accuracy by incorporating more case studies and data.
16. Write detailed about assigning economic value to oxygen by-product of water electrolysis to improve the business case.
17. Include all the equations, such as those for GHG emission calculation, LCA, and hydrogen production cost.

Future hydrogen economies imply environmental trade-offs and a supply-demand mismatch

Response letter

Tom Terlouw^{a,b,c*}, Lorenzo Rosa^d, Christian Bauer^c, and Russell McKenna^{b,c*}

^a Separation Processes Laboratory, Institute of Energy and Process Engineering, ETH Zurich, Zurich 8092, Switzerland

^b Chair of Energy Systems Analysis, Institute of Energy and Process Engineering, ETH Zurich, Zurich 8092, Switzerland

^c Technology Assessment Group, Laboratory for Energy Systems Analysis, 5232 Villigen PSI, Switzerland

^d Department of Global Ecology, Carnegie Institution for Science, Stanford, 94035, CA, United States of America

* Corresponding author: tterlouw@ethz.ch and russell.mckenna@psi.ch.

Dear editor and reviewers of the manuscript “*Future hydrogen economies imply environmental trade-offs and a supply-demand mismatch*” with ID: **NCOMMS-23-61509**. We thank you for your constructive comments and time. We confirm that we have addressed each comment in the revised manuscript as outlined below. Please find our answers to your comments in blue. Changes in the manuscript are shown in *italics*.

1 Reviewer 1:

The authors address some of the most pressing questions regarding the potential and implications of hydrogen as an energy carrier in the context of decarbonization. They highlight in particular: the challenge of water and land scarcity, life-cycle emissions of hydrogen production and issues or material scarcity. The authors also touch on other issues, such as the risks of hydrogen leakage. While these concerns are widely discussed in the literature, there is still a scarcity of rigorous, quantitative studies on these issues and what they might imply. In this vein, this paper is a welcome contribution, as it sets out to provide such a quantitative assessment. However, in my view, the paper is ultimately over-ambitious in trying to provide a meaningful quantitative contribution to all the different dimensions listed above. As a result, the paper does not address a lot of important questions regarding assumptions, data, sensitivities, etc. in detail. For instance, it is not clear how the authors arrive at the substantial GHG emissions from renewable hydrogen production. There is a reference to Life-cycle inventories. However, there is no discussion of these inventories and how they may change

in the context of decarbonization. Similarly, impacts on water are addressed, but implications and options - such as desalinization - are not addressed in detail. As a result, the paper does not provide substantial new insights into the challenges of scaling-up hydrogen production, but rather restates issues that are well-known in the debate. In my view, the paper could make a much more substantial contribution if the authors would choose to focus on one of the issues in more detail and provide a more nuanced discussion of the specific issue, including the specific design of the assessment, data uses, sensitivities and uncertainties in that area (ideally without having to go to the supplemental material). This way readers could obtain detailed insights into the state of current knowledge in the chosen field and the authors could provide a nuanced discussion of implications. Given the various different topics the authors attempt to address, this is not possible. Hence, the value of the paper is substantially lower..

Thank you for your constructive feedback. The main strength and uniqueness of our paper is that it provides a comprehensive, consistent, quantitative, spatially explicit, and future-oriented assessment of a range of hydrogen production technologies and their relevant environmental impacts and costs, which are likely to influence future hydrogen economies. It is entirely correct that most of these issues have been analyzed in the literature, but in a fragmented way, which does not allow for a consistent and meaningful evaluation of low-carbon hydrogen production at scale in terms of co-benefits and potential trade-offs. Such trade-offs have been illustrated in a newly generated Figure R1 below (Supplementary Figure A32). This figure clearly shows that there are trade-offs, for example, between countries with water scarcity in combination with large solar PV energy sources (Morocco) and countries with abundant wind energy sources with higher critical material utilization (Canada and Denmark).

In contrast, using results from individual analyses in the literature for such an evaluation would be prone to inconsistencies regarding, for example, system boundaries, input data and assumptions and thus be much less meaningful, potentially even misleading. A comprehensive and holistic analysis, such as ours, has not been performed in the literature yet. However, we do understand your concern very well and have provided a more detailed explanation regarding life cycle inventory and cost data used (see supplementary material C and D), and uncertainties of energy scenarios and further aspects (see Discussion).

In the following, you can find our point-by-point improvements to consider your main comments on the (i) explanation of GHG emissions of low-carbon hydrogen production pathways now and in the future, (ii) discussion on life cycle inventories, (iii) impacts on water scarcity and potential solutions, and (iv) additional uncertainties.

Page 4 of the manuscript: *(i) Hydrogen production via water electrolysis using solar PV electricity might result in substantial emissions in the short term—due to current PV wafer production in China with fossil energy—if*

Least country-specific cost-configurations (2050, 2°C) exhibit trade-offs between water, land, GHGs, materials, and costs.

Figure R1: Trade-offs between least cost configurations (2050, 2°C) of selected countries.

the capacity factor of the solar PV system is low (5.5–9.6 kg CO₂-eq. kg⁻¹ H₂). However, it can be reduced using ground-mounted solar PV systems as they have smaller embodied emissions from production (4.8–8.5 kg CO₂-eq. kg⁻¹ H₂). These embodied life cycle GHG emissions of renewables are presented in Table R1. In contrast, hydrogen production via water electrolysis with wind turbines has substantially lower emissions (2.3–3.8

kg CO₂-eq. kg⁻¹ H₂) due to a higher capacity factors and lower embodied emissions per unit of electricity needed for the electrolyzer. Our future scenarios reveal, however, that emissions from water electrolysis are likely to be reduced due to improved technology efficiencies and the overall decarbonization of the energy system. Modifying the electricity, cement, steel, and fuels sector in the background LCA database (using premise¹) results in a reduction to 1.4–3.4 kg CO₂-eq. kg⁻¹ H₂ and to 0.8–1.8 kg CO₂-eq. kg⁻¹ H₂ for hydrogen production via water electrolysis using solar PV and wind energy sources, respectively.

Table R1: Life cycle GHG emissions of renewable power sources for the different scenarios. Future emission factors are generated using premise¹ by modifying all available sectors. This table illustrates the emission factor of renewables decreases with more ambitious climate pathways and with higher capacity factors.

Electricity source	Residential			Ground-mounted			Onshore			Offshore			Unit
	Solar PV			Solar PV			Wind			Wind			
Capacity factor	0.125	0.175	0.225	0.125	0.175	0.225	0.25	0.35	0.45	0.35	0.45	0.55	[-]
Reference	0.069	0.049	0.038	0.049	0.035	0.027	0.015	0.010	0.008	0.014	0.011	0.009	[kg CO ₂ -eq./kWh]
BAU (2050)	0.021	0.015	0.012	0.020	0.014	0.011	0.009	0.006	0.005	0.009	0.007	0.006	[kg CO ₂ -eq./kWh]
2°C (2050)	0.013	0.010	0.007	0.013	0.009	0.007	0.006	0.004	0.003	0.007	0.005	0.004	[kg CO ₂ -eq./kWh]
1.5°C (2050)	0.011	0.008	0.006	0.011	0.008	0.006	0.005	0.003	0.003	0.005	0.004	0.003	[kg CO ₂ -eq./kWh]

Page 11 of the supplementary information: (ii) Ecoinvent data is used for LCA of wind turbines, ground-mounted solar PV, and residential solar PV (mounted on the roof). Most of the inventories are obtained from the open-source Python package premise (v.1.8.1)¹. These life cycle inventories are obtained from papers presented in Refs.^{2–5} Water consumption for water electrolysis is assumed to be 24 kg H₂O/kg H₂, considering the stoichiometric water requirements (9 kg H₂O/kg H₂ required while 8 kg O₂/kg H₂ is produced), and water losses due to water treatment (15 kg H₂O/kg H₂)⁶.

In our geospatial analysis, the autothermal reforming (ATR)+carbon capture and storage (CCS) dataset is used to represent natural gas reforming+CCS as the ATR dataset better represents low-carbon blue hydrogen production with higher capture efficiencies of CO₂ than conventional steam methane reforming (SMR)². Further, the ecoinvent database includes an activity for reverse osmosis. However, this activity is updated with recent environmental flow factors found in Ref.⁷, which shows that reverse osmosis using seawater has 117–177% higher environmental flows compared to the ecoinvent activity and is considered to be more reliable.

Water consumption impacts from ecoinvent and premise are usually less reliable than other environmental exchanges, and are therefore validated and corrected based on data from Refs.^{8,9}. For example, the difference between SMR and ATR with and without CCS has been considered using data from Ref.⁸ while coal gasification and wood gasification water consumption has been corrected based on recent data from Ref.^{8,9}.

Page 17 of the manuscript: (iii) Water desalination could be a promising technology to purify seawater at locations with access to brackish water, and is associated with marginal GHG emissions¹⁰ and a cost of less than 0.02 € kg⁻¹ H₂^{10,11}, which typically represents less than 1% of hydrogen production cost^{10,11}. However,

appropriate brine disposal in marine environments remains critical, as it can still impose significant environmental burdens^{11,12}. Additionally, reusing and cleaning wastewater could be another option to reduce water scarcity, although PEM electrolyzers require very high water purity. Deploying large-scale offshore hydrogen production platforms might be a promising way to reduce potential onshore land and water scarcity from large-scale hydrogen production.

Finally, additional sensitivity analyses have been performed on land utilization factors (provided in the supplementary information) to show uncertainty regarding optimal hydrogen production locations, the hydrogen supply curve, and the implication on hydrogen trade. These results are illustrated in Figures R2–R3, implying that with high land utilization factors, hydrogen production will be more concentrated and more hydrogen trade is required. Further discussion on the main uncertainties has also been included in the discussion section.

Page 15 of the manuscript: *(iv) Our hydrogen scenarios are inherently uncertain due to potential technology breakthroughs and policy development until 2050, which we considered by using four future scenarios in 2050 derived from IAM pathways and IRENA. Here, we focused on low-carbon hydrogen production mainly via PEM water electrolysis using wind and solar PV as power sources. However, due to the development of alternative electrolyzers and energy sources as well as policy, current low-carbon hydrogen production energy systems might be replaced by more cost-efficient pathways, although they are likely to exhibit other environmental trade-offs. Nevertheless, our analysis reveals that novel hydrogen production pathways should be assessed using global scope to identify potential environmental trade-offs and limitations.*

Figure R2: Land utilization factor of max. 1%. **a,c,e,g** Cost supply curves in year 2050 for business-as-usual, 2°C, 1.5°C, and 1.5°C (IRENA), respectively. **b,d,f,h** Selected economical locations for business-as-usual, 2°C, 1.5°C, and 1.5°C (IRENA), respectively. **j-k** Net H₂ supply for 2°C and 1.5°C, respectively.

Figure R3: Land utilization factor of max. 4%. **a,c,e,g** Cost supply curves in year 2050 for business-as-usual, 2°C, 1.5°C, and 1.5°C (IRENA), respectively. **b,d,f,h** Selected economical locations for business-as-usual, 2°C, 1.5°C, and 1.5°C (IRENA), respectively. **j-k** Net H₂ supply for 2°C and 1.5°C, respectively.

2 Reviewer 2:

The authors provided an extensive economic-environmental assessment of PV- and wind-based green hydrogen production considering multiple scenarios of demand in the year 2050. The manuscript is very well written, and the results are thoughtfully analyzed. This study should be valuable to many stakeholders and decision-makers on different levels. I have no significant comments on its content. Hence, I believe it can be published as is.

Our sincere gratitude for your positive response and review of our work. We are happy to learn that you think that our work is most likely valuable for different stakeholders and decision-makers.

3 Reviewer 3:

Manuscript Title: Future hydrogen economies imply environmental trade-offs and a supply-demand mismatch.

The authors carried out an in-depth survey on the critical aspects of future hydrogen economies, emphasizing the need to replace current fossil fuel-based hydrogen production with low-carbon alternatives for effective decarbonization. The study quantifies the costs and environmental impacts associated with large-scale hydrogen economies, projecting demand scenarios for 2050. The paper emphasizes the necessity of a comprehensive understanding, considering various production technologies, future scenarios, and both cost and environmental factors. I recommend this article with minor revisions.

The manuscript is written clearly; however, the following points should be clarified or added.

Thank you for your positive response and for taking the time for your comprehensive review and suggestions for improvement.

1. Non-linear trend correlations have critical limitations, including the impact of outliers, context specificity, hidden variables, and attractor reconstruction in time series data. These limitations need to be justified. Additionally, the manuscript should provide a more comprehensive explanation of the "novel non-linear trend correlations" method, highlighting its advantages over alternative approaches.

Thank you for this remark. In our work, one important complexity is the very large number of locations for which the electrolysis-based hydrogen production systems have to be optimally designed using a global scope. Thus, it is hardly feasible to design all of them individually by solving each individual optimization problem. To determine the optimal design in an approximate way, we use non-linear curve fittings to quan-

tify the following location-specific aspects: (i) the land share available to install onshore wind and solar PV, (ii) the battery energy storage capacity, and (iii) the electrolyzer capacity. Indeed, different methods exist to address the complexity of optimally designing hydrogen production systems using global scope. Non-linear curve fittings are an excellent approach due to their ability to capture non-linear relationships, offer interpretable coefficients, and allow for applicability and reuse for researchers. In this way, non-linear curve fittings provide a reasonable balance between computational feasibility and the need to estimate the optimal design at a grid pixel level. However, they also come with drawbacks, such as neglecting outliers.

Another interesting option would have been clustering locations, for example, based on weather conditions, renewable energy potential, intermittency, and land use. The drawbacks and benefits, and a short review of non-linear trend estimations, as opposed to clustering, are provided in the supplementary information (Page 4 of the supplementary information) and are shortly mentioned in the discussion:

A curve-fitting approach is used to estimate the optimal design of large-scale hydrogen production facilities. More specifically, non-linear curve fittings are used to quantify the following location-specific aspects: (i) the land share available to install onshore wind and solar PV, (ii) the battery energy storage capacity, and (iii) the electrolyzer capacity. Indeed, different methods exist to address the complexity of optimally designing hydrogen production systems using global scope.

Non-linear curve fittings are a reasonable approach due to their ability to capture non-linear correlations and offer interpretable coefficients, which allow researchers to reuse them. In this way, non-linear curve fittings provide an appropriate balance between computational complexity and the need to determine the (near-)optimal design at each geospatial grid cell (approximately 25 km × 25 km). On the one hand, non-linear curve fittings are typically limited to capturing data outliers, which poses challenges in representing the actual complexity of the system. On the other hand, our analysis does not focus on designing case studies but looks at the global perspective, which makes outliers of individual system designs less important. Clustering might represent an alternative; for example, locations could be clustered based on weather conditions, renewable energy potential, intermittency, and land use. However, determining the optimal number of clusters is complex and likely requires many clusters to capture the set of location-specific aspects globally, potentially leading to an oversimplification of variability within clusters by assuming homogeneity.

2. The scenarios considered for global scale-up of hydrogen production are well-defined. However, a more extensive discussion on the rationale behind choosing these specific scenarios, including the underlying assumptions and potential variations, would add depth to the research.

Thank you for this suggestion. We have expanded the information on the different future energy scenarios

considered in the Methods section, page 21:

Here, we consider a broad space of potential development trajectories. Therefore, various shared socioeconomic pathway 2 (SSP2) scenarios—based on the outputs of integrated assessment models (IAMs)—are chosen since they represent middle-of-the-road pathways to socio-economic (intermediate) challenges for adaptation and mitigation^{13,14}. As such, SSP2 represents a pathway characterized by moderate challenges and uncertainties, which ensures a balanced scenario between potential extreme future development trajectories.

- *Reference (2022): representing the current global economy. For this purpose, the current costs of technologies are applied and the ecoinvent 3.9.1 database is used as background LCI. For the current situation, an overall production of 90 Mt H₂ yr⁻¹ has been considered.*
- *Business-as-usual (BAU): representing a possible future global economy in 2050 using the current socioeconomic trend, i.e., a business-as-usual pathway that extrapolates historical developments. For this purpose, the background LCA database is modified using the SSP2-Base scenario from REMIND^{15,16}. This pathway of REMIND corresponds to an overall production of 111 Mt H₂ yr⁻¹.*
- *2°C: representing a future global economy in 2050 that limits global warming to 2°C. For this purpose, the background LCA database is modified using the SSP2-PkBudg1150 scenario from REMIND, which corresponds to an overall production of 451 Mt H₂ yr⁻¹^{15,16}.*
- *1.5°C: representing a future global economy in 2050 that limits global warming to 1.5°C. For this purpose, the background LCA database is modified using the SSP2-PkBudg500 scenario from REMIND, which corresponds to an overall production of 364 Mt H₂ yr⁻¹^{15,16}.*
- *1.5°C (IRENA): representing a future global economy in 2050 that limits global warming to 1.5°C. For this purpose, the background LCA database is modified using SSP2-PkBudg500 scenario from REMIND, but corresponding to an overall production of 614 Mt H₂ yr⁻¹ as described in the IRENA scenario¹⁷. The 1.5°C (IRENA) pathway has substantial hydrogen integration due to the assumption that hydrogen will be produced in least-cost locations and transported—likely converted to liquid hydrogen, ammonia, or synthetic fuels—to demand centres.*

The chosen future scenarios differ substantially regarding overall hydrogen demand in 2050. This demand is, among other factors, mainly driven by climate goals, assumed technology developments and degree of direct electrification of energy end-use application. Here, 1.5°C (IRENA) likely represents the “upper level” of future hydrogen demand, while the BAU scenario exhibits the “lower level” of future hydrogen demand. Reality will be likely in between these pathways, potentially represented by the other 1.5°C and 2°C scenarios.

3. It is important for the authors to discuss the uncertainty in the impacts of climate change, especially regarding blue hydrogen. Including a more detailed discussion on how various other hydrogen production methods might affect climate change would offer a more comprehensive perspective, providing strong justification for choosing green hydrogen for a global study.

Thank you for this remark. We already briefly touched upon the impact of blue hydrogen in the description of the evaluation framework and scenarios, but we do agree that we could have a more elaborate description to justify focusing on green hydrogen by showing the disadvantages of alternative low-carbon hydrogen pathways. Therefore, we have expanded this section and introduced the following sentences on page 3 of the manuscript:

Further, some other potentially prominent low-carbon hydrogen production pathways exhibit ambiguous climate change impacts, especially blue hydrogen^{18,19}. Blue hydrogen is hydrogen production via steam methane reforming combined with CCS, but the climate impacts are uncertain due to potential fugitive methane emissions, the capture rate of CCS, and the global warming potential metric applied. Thus, blue hydrogen is often considered as 'bridging technology' filling a gap in low-carbon hydrogen demand as long as electrolysis would not be sufficiently scaled up^{18,20}.

4. The author mentioned environmental trade-offs, but a more elaborate discussion, especially regarding the considerable potential of wind energy, is crucial for understanding the true environmental footprint of each method.

Thank you for this comment. Indeed, onshore and offshore wind will likely be an important element of a future low-carbon hydrogen supply system. In the new version of the manuscript (Results, page 11), we further elaborate on the influence of wind capacity installations on the overall impacts.

The installed power generation technologies slightly differ between the different pathways. However, most simulated hydrogen economies with substantial electrolytic hydrogen production require several terawatts of electrolyzers, onshore and offshore wind, and solar PV—up to 4.7 TW for electrolyzers in 1.5°C scenarios (IRENA). In general, more onshore wind is installed as opposed to solar PV and offshore wind, mainly due to their lower levelized costs of electricity as a result of lower expected capital expenditures and higher electricity yields. The implications of installing large amounts of onshore wind (and other renewables) extend beyond economic and environmental aspects but also influence non-technical-aspects, such as the energy landscape (i.e., their scenicness), communities (local resistance), and regulatory frameworks²¹.

5. The authors should discuss the potential impact of future trends, such as advancements in technology and changes in policy, on the presented results.

Thank you for this remark. Indeed, our results are specific for future energy scenarios and underlying assumptions but are inherently uncertain. However, it is worth noting that our four scenarios include already a broad range of future developments in terms of policies and technology development (REMIND BAU, 2°C, and 1.5°C scenarios, and IRENA 1.5°C). We further discuss this point now in the discussion section on page 11 of the manuscript:

Our hydrogen scenarios, derived from IAM pathways and IRENA, are inherently uncertain due to potential technology breakthroughs and policy development until 2050. Here, we focused on low-carbon hydrogen production mainly via PEM water electrolysis using wind and solar PV as power sources. However, due to the development of alternative electrolyzers and energy sources as well as policy, current low-carbon hydrogen production energy systems might be replaced by more cost-efficient pathways, although they are likely to exhibit other environmental trade-offs. Nevertheless, our analysis reveals that novel hydrogen production pathways should be assessed using global scope to identify potential environmental trade-offs and limitations.

6. More detailed information about the data sources and modelling assumptions used in the analysis should be included.

Thank you for this suggestion. We have included additional explanation about life cycle inventory data used, which is provided in supplementary information C (page 11). Further, an explanation about cost data is updated and given in supplementary information D.

Ecoinvent data is used for wind turbines, ground-mounted solar PV, and residential solar PV (mounted on the roof). Most of the inventories are obtained from open-source Python package premise (v.1.8.1)¹. These life cycle inventories are obtained from papers presented in Refs.^{2–5} Water consumption for water electrolysis is assumed to be 24 kg H₂O/kg H₂, considering the stoichiometric water requirements (9 kg H₂O/kg H₂ required while 8 kg O₂/kg H₂ is produced), and water losses due to water treatment (15 kg H₂O/kg H₂)⁶.

In our geospatial analysis, the autothermal reforming (ATR)+carbon capture and storage (CCS) dataset is used to represent natural gas reforming+CCS as the ATR dataset better represents low-carbon blue hydrogen production with higher capture efficiencies of CO₂ than conventional steam methane reforming (SMR)². Further, the ecoinvent database includes an activity for reverse osmosis. However, this activity is updated with recent environmental flow factors found in Ref.⁷, which shows that reverse osmosis using seawater has 117–177% higher environmental flows compared to the ecoinvent activity and is considered to be more reliable.

Water consumption impacts from ecoinvent and premise are usually less reliable than other environmental ex-

changes, and are therefore validated and corrected based on data from Refs.^{8,9}. For example, the difference between SMR and ATR with and without CCS has been considered using data from Ref.⁸ while coal gasification and wood gasification water consumption has been corrected based on recent data from Ref.^{8,9}.

7. Explain the specific methods used for cost and GHG emission calculations in different regions.

Thank you for this suggestion. We use the same cost metric to calculate the levelized cost of hydrogen, which is grid cell-specific. The equation of hydrogen production cost is now provided in the Methods section (page 22). Further, location-specific environmental burdens are calculated by first calculating technology-specific LCA impacts and multiplying them by the specific technology capacity considering the technology lifetime. Next, they are aggregated for the entire hydrogen production system per grid cell. We have introduced a section called ‘Main performance indicators’ in the Methods section to introduce Equations 1–4 on page 22 of the manuscript:

Main performance indicators

The main economic performance indicator used is the levelized cost of hydrogen production (C_{H2}) per grid cell, which is calculated by considering annualized investments ($C_{inv,an}$), operation & maintenance costs (C_{op}), and fixed operation & maintenance costs (C_{om}), divided by the annual hydrogen production rate in a grid cell ($H_{2,total}$)¹⁰.

$$C_{inv,an} = \sum_{i=1}^M \frac{\gamma (1 + \gamma)^L}{(1 + \gamma)^L - 1} C_{inv}^i, \quad (1)$$

$$C_{H2} = \frac{C_{op} + C_{inv,an} + C_{om}}{H_{2,total}}, \quad (2)$$

where γ is the WACC [–], the set of technologies (\mathcal{M}) in the hydrogen production system is indexed with $i \in \{1, 2, \dots, M\}$, L is the lifetime of the hydrogen production system. The lifetime of the hydrogen production system is set to 30 years.

The following Equations (3) and (4) express the environmental impacts of the operation (G_{op} , such as hydrogen leakages) and embodied emissions (G^i) of energy technologies, and the overall environmental impact of hydrogen production per kilogram of hydrogen produced per grid cell (G_{H2}), respectively. It is important to highlight that G_{inst} encompass the impact of an environmental impact category, and can thus be calculated separately for GHG emissions, land occupation, and the materials needed for manufacturing and replacing an energy technology. Thus, for each energy technology, the LCA impacts are calculated and multiplied by their installed capacities

divided by their lifetime and/or capacity factor²².

$$G_{inst} = \frac{\sum_{i=1}^M G^i \frac{L}{L^i}}{L}, \quad (3)$$

$$G_{H2} = \frac{G_{inst} + G_{op}}{H_{2,total}}, \quad (4)$$

where L^i is the lifetime of technology i [year].

The life cycle inventory used (from background LCA database ecoinvent 3.9.1²³ using system model “cut-off by classification”) to calculate LCA impacts is presented in supplementary Table A1.

8. Discuss potential limitations of the analysis, such as uncertainties in future scenarios and data availability.

Thank you for this remark. We believe that this comment is similar to comment 5. Thus, we have included additional discussion on the uncertainty of future energy scenarios and cost and life cycle inventory data of hydrogen production pathways.

9. Discuss the potential for green hydrogen trade between regions with abundant resources and those with high demand.

Thank you for this suggestion. We have elaborated on potential hydrogen trade in the results section on page 8 of the manuscript. Please find the additional explanation below:

The two subplots at the bottom of Figure 3 show the geographical mismatch between hydrogen production and demand based on the 2°C (451 Mt H₂ yr⁻¹) and 1.5°C (364 Mt H₂ yr⁻¹) scenarios from REMIND. Dark green colors illustrate an excess production of hydrogen while dark red colors indicate shortages of hydrogen supply indicating import demand. For example, this is illustrated by local hydrogen supply shortages in India and South-East Asia, and an excess production of hydrogen in Canada, the USA, and Australia. India and South-East Asia have a deficit of low-cost hydrogen production locations due to their less suitable climate and land availability and high population densities. On the contrary, Canada, the USA, and Australia are the only continents with sufficient economical hydrogen production, implying that the conversion to hydrogen (carriers) and transportation is inevitable for of a future low-carbon economy if large-scale hydrogen economies are established. Importantly, this requires the development of a dedicated hydrogen transportation network, which has not been analyzed in this work but would require additional investments and lead to additional impacts (see Discussion).

10. Explore technological advancements that could mitigate water scarcity issues in hydrogen production (e.g., seawater desalination, wastewater reuse).

Thank you for this remark. We agree that technological advancement could mitigate water scarcity issues. This has been discussed and further elaborated on in the Discussion on page 17:

Water desalination could be a promising technology to purify seawater at locations with access to brackish water; and is associated with marginal GHG emissions¹⁰ and a cost of less than 0.02 € kg⁻¹ H₂^{10,11}, which typically represents less than 1% of hydrogen production cost^{10,11}. However, appropriate brine disposal in marine environments remains critical, as it can still impose significant environmental burdens^{11,12}. Additionally, reusing and cleaning wastewater could be another option to reduce water scarcity, although PEM electrolyzers require very high water purity. Deploying large-scale offshore hydrogen production platforms might be a promising way to reduce potential onshore land and water scarcity from large-scale hydrogen production.

11. Including land transformation for biomass production significantly increases land use requirements. This should be highlighted as a potential limiting factor for countries with large hydrogen ambitions.

Thank you for this comment. We have highlighted this in the Discussion and Conclusion section (page 16) of the new version of the manuscript.

This figure increases significantly by almost one million km² if we include the land transformation needed for biomass-based hydrogen production in a 1.5°C scenario of REMIND. Thus, land occupation could be another limiting factor for countries with large-scale hydrogen production ambitions via biomass sources.

12. The concept of limiting land utilization to 2% seems arbitrary. Justifying this limit based on specific land cover data would strengthen the argument.

Thank you for this suggestion. First, it is worth noting that we determine the amount of land available based on the current land cover data and multiply it with land factors considering the land use type. Second, we multiply this total land area with a land utilization factor to obtain more realistic land availability. Without such a factor, this would lead to the installation of huge large-scale hydrogen production facilities in several global locations, which would be unrealistic. However, as you mentioned, the 2% factor is an arbitrary choice, although such land use factors have been previously applied in previous works⁶ and in policy documents. To illustrate this, a 2% land utilization factor has been used as a goal in Germany for onshore wind installations²⁴ and about 1% of current global land use is used for urban areas²⁵. To consider a wider range of land utilization factors, we provide two sets of additional results applying 1% and 4% of land utilization

factors in Figures R2–R3 (please refer to page 6 and 7 of this response letter) and in Supplementary Information F1. These figures illustrate that with high land utilization factors, hydrogen production will be more concentrated and a larger hydrogen trade is required. In addition, with low land utilization factors, it will be difficult to achieve high hydrogen production quotas, and many geographical areas must be subjected to smaller hydrogen production facilities.

13. The criticality of specific materials, particularly iridium and dysprosium, is well-discussed. Expanding on potential mitigation strategies, such as material substitution, recycling, or improved efficiency, would be helpful.

Thank you for this suggestion. We have discussed this at the end of the results section, page 13. We have expanded and modified it based on your remark.

It is, however, unlikely that the total extraction of materials occurs in a single year. Further, material criticality might be reduced by improved material efficiency, recycling of metals and materials, annual production volumes can be increased, and alternative technologies or material substitutes might be developed²⁶. These factors could collectively decrease the potential criticality of lithium and other materials by 2050.

14. The paper could benefit from a more thorough comparison with relevant existing studies, highlighting the unique contributions and potential areas for further research.

Thank you for this suggestion. We have briefly discussed previous research in the introduction section. However, we agree that it would be useful to have a more thorough comparison of existing literature and the current limitations of them. This has been now introduced on page 8 of the Supplementary Information.

The main body of the article discussed the current limitations of literature on the potential upscaling towards large-scale hydrogen economies. Here, a couple of recent studies on large-scale hydrogen production are discussed. For example, Terlouw et al.¹⁰ provided a comprehensive environmental and techno-economic assessment of optimized hydrogen production now and in the future for five case studies on geographical islands in Europe. These geographical islands exhibit beneficial hydrogen production potential and are used in an analysis upscaling towards 9.5 EJ (per year) hydrogen production via water electrolysis using the impacts of these case studies. Hence, they excluded a comprehensive geospatial quantification of location-specific aspects, except for the five case studies, thus, not considering other global regions for hydrogen production potential. A recent study by Weidmer et al.²⁷ determines the environmental sustainability of future hydrogen economies in 2050 of 500 Mt/hydrogen economies by considering different hydrogen production pathways including grey, blue, and green hydrogen. The impacts are compared to planetary boundaries using prospective LCA. While the authors mention

that location-specific impacts are crucial, case studies for hydrogen production pathways were used to derive to the impacts of prospective large-scale hydrogen production quotas. Finally, Tonelli et al.⁶, incorporated water and land considerations to explore the limitations of a future global hydrogen economy solely focusing on electrolytic hydrogen production using geospatial analysis but considering one socio-economic narrative and excluding the optimal sizing, cost, and other environmental factors.

Thus, prior analyses have been limited by exploring a single socio-economic narrative, while we argue that future hydrogen production scenarios are highly uncertain and depend on socio-economic development pathways. Given the diverse range of possible development scenarios, there is an urgent need for comprehensive assessments that address such uncertainties to determine the potential global environmental implications of future hydrogen production economies. Additionally, there are still environmental burdens and trade-offs that have not been evaluated, such as material utilization and overall trade-offs of hydrogen production pathways. These omissions impede a comprehensive understanding of the potential challenges and opportunities of transitioning to a hydrogen-based economy. Lastly, prior global geospatial analyses have often overlooked the optimal sizing of hybrid hydrogen production systems—considering both regional solar PV and wind potentials—or have done so in a simplified way. Besides, these analyses frequently—in our view unreasonably—assumed the existence of a local power grid in remote regions. Such simplifications typically lead to either under- or over-dimensioning hydrogen production systems and neglecting the consideration of more complex (off-grid) hybrid energy systems with flexibility options, such as batteries and renewable electricity curtailment. We argue that these systems should be designed in a (near-)optimal way using optimization techniques to minimize cost, and to better account for curtailment and battery electricity storage^{10,22,28,29}.

15. The authors should refine land use factors and curve-fit accuracy by incorporating more case studies and data.

Thank you for this remark. We have checked the land use factors and refined our curve-fits by increasing the number of case studies from 50 to 100 per energy scenario, resulting in a total of 400 optimizations performed to establish the curve-fits. The new curve-fits and plot of locations selected can be found in supplementary information A.

16. Write detailed about assigning economic value to oxygen by-product of water electrolysis to improve the business case.

Thank you for this suggestion. We have decided to apply a conservative approach to oxygen value and, therefore, do not consider the value of oxygen. We focus on off-grid hydrogen production systems, and the current demand and supply chain of oxygen is not developed. However, we do agree that in certain regions, the oxygen prices are higher, for example, due to the close vicinity of a hospital, and that there could be

substantial economic value given to oxygen. We discuss this in the discussion section (page 16) in the new version of the manuscript:

Similarly, no economic value has been given to oxygen as a by-product of hydrogen production since we focus on off-grid hydrogen production systems. A large-scale demand and supply chain of oxygen is not developed in most geographical regions (except in cities near hospitals), especially in off-grid regions. However, oxygen can have a substantial economic value between a large range 0.04³⁰–3³¹ euro/kg H₂, where the upper level represents oxygen utilisation in hospitals although the high oxygen medical standards might reduce the economic benefits of using oxygen as a by-product. Thus, considering oxygen as a useful by-product could result in substantial revenues for hydrogen production. And it could also reduce its LCA-based greenhouse gas emissions, especially when applying economic allocation given that 8 kg of O₂ is produced per kilogram of hydrogen production.

17. Include all the equations, such as those for GHG emission calculation, LCA, and hydrogen production cost.

Thank you for this comment. We believe this comment is similar to considering comment 7, which we addressed by describing how levelized hydrogen production cost and environmental impacts are calculated by introducing a new section in the methods called “Main performance indicators” (page 22 of the manuscript).

References

1. Sacchi, R. *et al.* PROspective EnvironMental Impact asSEment (premise): A streamlined approach to producing databases for prospective life cycle assessment using integrated assessment models. *Renewable and Sustainable Energy Reviews* **160**, 112311 (2022).
2. Antonini, C. *et al.* Hydrogen production from natural gas and biomethane with carbon capture and storage - A techno-environmental analysis. *Sustainable Energy & Fuels* **4**, 2967–2986 (2020).
3. Wokaun, A. & Wilhelm, E. *Transition to hydrogen: pathways toward clean transportation* (Cambridge University Press, 2011).
4. Li, J., Wei, Y.-M., Liu, L., Li, X. & Yan, R. The carbon footprint and cost of coal-based hydrogen production with and without carbon capture and storage technology in China. *Journal of Cleaner Production* **362**, 132514 (2022).
5. Gerloff, N. Comparative Life-Cycle-Assessment analysis of three major water electrolysis technologies while applying various energy scenarios for a greener hydrogen production. *Journal of Energy Storage* **43**, 102759 (2021).
6. Tonelli, D. *et al.* Global land and water limits to electrolytic hydrogen production using wind and solar resources. *Nature Communications*, **14**, 5532 (2023).
7. Fayyaz, S. *et al.* Life cycle assessment of reverse osmosis for high-salinity seawater desalination process: Potable and industrial water production. *Journal of Cleaner Production* **382**, 135299 (2023).
8. (PI), A. E. *et al.* *Life-Cycle Analysis of Water Consumption for Hydrogen Production* 2016. https://www.hydrogen.energy.gov/pdfs/review16/sa039_elgowainy_2016_o.pdf (2023).
9. (NETL), N. E. T. L. *COMPARISON OF COMMERCIAL, STATE-OF-THE-ART, FOSSIL-BASED HYDROGEN PRODUCTION TECHNOLOGIES* Nov. 19, 2023. <https://netl.doe.gov/projects/files/ComparisonofCommercialSta041222.pdf> (2023).
10. Terlouw, T., Bauer, C., McKenna, R. & Mazzotti, M. Large-scale hydrogen production via water electrolysis: a techno-economic and environmental assessment. *Energy & Environmental Science* **15**, 3583–3602 (2022).
11. Khan, M. *et al.* Seawater electrolysis for hydrogen production: a solution looking for a problem? *Energy & Environmental Science* **14**, 4831–4839 (2021).
12. Panagopoulos, A. & Haralambous, K.-J. Environmental impacts of desalination and brine treatment- Challenges and mitigation measures. *Marine Pollution Bulletin* **161**, 111773 (2020).
13. Koch, J. & Leimbach, M. SSP economic growth projections: Major changes of key drivers in integrated assessment modelling. *Ecological Economics* **206**, 107751 (2023).

14. Riahi, K. *et al.* The Shared Socioeconomic Pathways and their energy, land use, and greenhouse gas emissions implications: An overview. *Global environmental change* **42**, 153–168 (2017).
15. Luderer, Gunnar and Leimbach, Marian and Bauer, Nico and Kriegler, Elmar and Baumstark, Lavinia and Bertram, Christoph and Giannousakis, Anastasis and Hilaire, Jerome and Klein, David and Levesque, Antoine and Mouratiadou, Ioanna and Pehl, Michaja and Pietzcker, Robert and Piontek, Franziska and Roming, Niklas and Schultes, Anselm and Schwanitz, Valeria Jana and Strefler, Jessica. *Description of the REMIND model (Version 1.6)* 2015. <https://ssrn.com/abstract=2697070>.
16. Baumstark, L. *et al.* REMIND2.1: Transformation and innovation dynamics of the energy-economic system within climate and sustainability limits. *Geoscientific Model Development Discussions*, 1–50 (2021).
17. International Renewable Energy Agency, A. D. *Global hydrogen trade to meet the 1.5°C climate goal: Part I – Trade outlook for 2050 and way forward 2022*. https://www.irena.org/-/media/Files/IRENA/Agency/Publication/2022/Jul/IRENA_Global_hydrogen_trade_part_1_2022_.pdf.
18. Bauer, C. *et al.* On the climate impacts of blue hydrogen production. *Sustainable Energy & Fuels* **6**, 66–75 (2022).
19. Howarth, R. W. & Jacobson, M. Z. How green is blue hydrogen? *Energy Science & Engineering* **9**, 1676–1687 (2021).
20. Ueckerdt, F. *et al.* On the cost competitiveness of blue and green hydrogen. *Joule* **8**, 104–128 (2024).
21. McKenna, R. *et al.* High-resolution large-scale onshore wind energy assessments: A review of potential definitions, methodologies and future research needs. *Renewable Energy* **182**, 659–684 (2022).
22. Terlouw, T. *et al.* Optimal economic and environmental design of multi-energy systems. *Applied Energy* **347**, 121374 (2023).
23. Wernet, G. *et al.* The ecoinvent database version 3 (part I): overview and methodology. *The International Journal of Life Cycle Assessment* **21**, 1218–1230 (2016).
24. *Expanding wind energy for Germany — Federal Government* July 8, 2022. <https://www.bundesregierung.de/breg-de/schwerpunkte/klimaschutz/onshore-wind-energy-act-2060954> (2024).
25. Buchhorn, M. *et al.* Copernicus global land service: Land cover 100m: collection 3: epoch 2019: Globe. *Version V3.0.1* (2020).
26. Helbig, C., Schrijvers, D. & Hool, A. Selecting and prioritizing material resources by criticality assessments. *One Earth* **4**, 339–345 (2021).
27. Weidner, T., Tulus, V. & Guillén-Gosálbez, G. Environmental sustainability assessment of large-scale hydrogen production using prospective life cycle analysis. *international journal of hydrogen energy* **48**, 8310–8327 (2023).

28. Terlouw, T., AlSkaif, T., Bauer, C., Mazzotti, M. & McKenna, R. Designing residential energy systems considering prospective costs and life cycle GHG emissions. *Applied Energy* **331**, 120362 (2023).
29. Gabrielli, P., Gazzani, M., Martelli, E. & Mazzotti, M. Optimal design of multi-energy systems with seasonal storage. *Applied Energy* **219**, 408–424. ISSN: 0306-2619 (2018).
30. Dorris, C., Lu, E., Park, S. & Toro, F. *High-Purity Oxygen Production Using Mixed Ionic-Electronic Conducting Sorbents* Senior Design Reports (CBE) (Department of Chemical, Biomolecular Engineering School of Engineering, and Applied Science University of Pennsylvania, 2016).
31. Maggio, G., Squadrito, G. & Nicita, A. Hydrogen and medical oxygen by renewable energy based electrolysis: A green and economically viable route. *Applied Energy* **306**, 117993 (2022).

REVIEWER COMMENTS

Reviewer #1 (Remarks to the Author):

I am not in a position to review the details of the modelling and the data and assumptions that went into it. However, I am still not convinced that this comprehensive and therefore superficial discussion of the different dimensions of renewable hydrogen deployment adds significant new insights. Moreover, it does not become clear from the presentation what is driving the results.

It is also not clear from the analysis whether the analysis of emissions for the fossil-based technologies are subjected to similar system boundaries as the renewable hydrogen pathway. Having looked at the Antonini et al. (2020) paper, it seems for instance that the energy needs of CCS are not included in the assessment. Given the focus on a rigorous and extensive LCA approach when considering the emissions of renewable hydrogen, this seems odd. In my view, this paper needs to be broken down into more manageable steps before such a comprehensive assessment can be done with the rigour that is implied in the LCA of renewable hydrogen.

Reviewer #3 (Remarks to the Author):

The author's response to the critical comments are highly satisfactory.

Future hydrogen economies imply environmental trade-offs and a supply-demand mismatch

RESPONSE LETTER

Tom Terlouw^{a,b,c*}, Lorenzo Rosa^d, Christian Bauer^c, and Russell McKenna^{b,e*}

^a Separation Processes Laboratory, Institute of Energy and Process Engineering, ETH Zurich, Zurich 8092, Switzerland

^b Chair of Energy Systems Analysis, Institute of Energy and Process Engineering, ETH Zurich, Zurich 8092, Switzerland

^c Technology Assessment Group, Laboratory for Energy Systems Analysis, 5232 Villigen PSI, Switzerland

^d Department of Global Ecology, Carnegie Institution for Science, Stanford, 94035, CA, United States of America

^e Laboratory for Energy Systems Analysis, 5232 Villigen PSI, Switzerland

* Corresponding author: tom.terlouw@psi.ch and russell.mckenna@psi.ch.

Dear editor and reviewers of the manuscript “*Future hydrogen economies imply environmental trade-offs and a supply-demand mismatch*” with ID: **NCOMMS-23-61509A**. We thank you for your constructive comments and time. We confirm that we have addressed each comment (in **bold** font) in the revised manuscript as outlined below. Please find our answers to your comments in black. Changes in the manuscript are shown in blue.

Reviewer 1:

I am not in a position to review the details of the modelling and the data and assumptions that went into it. However, I am still not convinced that this comprehensive and therefore superficial discussion of the different dimensions of renewable hydrogen deployment adds significant new insights. Moreover, it does not become clear from the presentation what is driving the results. It is also not clear from the analysis whether the analysis of emissions for the fossil-based technologies are subjected to similar system boundaries as the renewable hydrogen pathway. Having looked at the Antonini et al. (2020) paper, it seems for instance that the energy needs of CCS are not included in the assessment. Given the focus on a rigorous and extensive LCA approach when considering the emissions of renewable hydrogen, this seems odd. In my view, this paper needs to be broken down into more manageable steps before such a comprehensive assessment can be done with the rigour

that is implied in the LCA of renewable hydrogen.

Thank you for your critical feedback. Based on your feedback, we have clarified the following concerns: (i) system boundaries of hydrogen production pathways, (ii) improved structure of the methods section, and (iii) main parameters driving the results.

(i) First of all, we want to make clear that energy requirements for carbon capture and storage (CCS) are included in the life cycle inventory of the study of Antonini et al. (2020)¹, thus, also in our work. Please find the following quotations in the latter reference:

“Higher CO₂ recovery and the addition of a low-temperature water gas shift further decrease the impacts on climate change, even if the electricity requirements increase, and even if this electricity is associated with rather high GHG emissions (ENTSO-E electricity mix with 0.42 kg CO₂-eq. per kW per h).” (p. 2981 of Ref.¹)

“However, SMR and ATR with CCS perform worse than without CCS regarding other environmental burdens as a result of increasing energy consumption and the (comparatively small) burdens associated with transport and storage of CO₂.” (p. 2982 bottom left of Ref.¹)

However, to clarify this and other assumptions, we have sketched (cradle-to-gate) system boundaries of the hydrogen production pathways to visualize the LCA modelling and assumptions for hydrogen production (at 25–30 bar). The sketches of the system boundaries of the non-water electrolysis-based hydrogen pathways are now explained and presented in Supplementary Note 1 (Pages 1–3). We have provided the following explanation:

Figure R1 illustrates the system boundaries of our product system with hydrogen production via SMR with (including the red blocks) and without CCS (excluding the red blocks). All processes are included, from cradle-to-gate up to hydrogen produced with a pressure of 25–30 bar, such as natural gas upgrading, transport, catalyst and adsorbent production, and capturing, transportation, and storage of CO₂ (in case of CCS).

Figure R2 shows the system boundaries of our product system with hydrogen production via biomass gasification with (including the red blocks) and without CCS (excluding the red blocks). All processes are included, from cradle-to-gate up to hydrogen produced via biomass gasification with a pressure of 25–30 bar, such as wood sawing, chipping, gasification unit, and capturing, transportation, and storage of CO₂ (in case of CCS).

Figure R1: System boundaries of hydrogen production *via* steam methane reforming (SMR) with and without carbon capture and storage (CCS), reproduced from Ref¹.

Figure R2: System boundaries of hydrogen production *via* biomass gasification with and without carbon capture and storage (CCS), reproduced from Ref².

Figure R3 visualizes the system boundaries of our product system with hydrogen production via coal gasification with (including the red blocks) and without CCS (excluding the red blocks). All processes are included, from cradle-to-gate up to hydrogen produced via coal gasification with a pressure of 25–30 bar, such as coal mining, preparation, gasification, and capturing, transportation, and storage of CO₂ (in case of CCS).

Figure R3: System boundaries of hydrogen production via coal gasification with and without carbon capture and storage (CCS).

We are confident that we have clarified the system boundaries and the LCA modelling of non-water-electrolysis pathways. It is worth noting that the system boundaries of the water electrolysis pathways are consistent and, thus, are also cradle-to-gate up to hydrogen with a pressure of 25–30 bar (outlet pressure of hydrogen from PEM electrolyzer) and are visualized in Figure 1 (of the main manuscript).

(ii) Second, we have improved the structure of the methods by restructuring it to the following:

- Evaluation framework and scope.
- System boundaries and life cycle analysis.
- Hybrid hydrogen production systems.
- Potential of renewables and land utilization.
- Techno-economic assumptions and scenarios.
- Main performance indicators.

To achieve this, we have moved the ‘Evaluation framework and scope’ to the Methods section to have the Introduction directly followed by the Results. We are confident that the new structure of the Methods reads better since it describes the evaluation framework and scope of this work first. Next, we discuss system boundaries and life cycle analysis, followed by hybrid hydrogen systems, the potential of renewables and land utilization, current and future hydrogen scenarios, and main performance indicators.

(iii) Third, we have performed an additional sensitivity analysis (considering all spatial grid cells globally) to determine influencing parameters for costs and GHG emissions, see Figure R4 (Supplementary Note 8). Based on our main results and sensitivity analyses, we have summarized the main factors influencing our results at the end of the results section (page 12):

In summary, the main factors driving our results are the future global macroeconomic development (e.g. influenced by technology development and learning), hydrogen demand (e.g. influenced by policy and collaboration), and potential resource constraints, such as material and land availability. For our global cost analysis of electrolytic hydrogen production, the capacity factor of hydrogen production (and renewables) has the highest impact (Supplementary Figure 36). The investments in electrolyzers and onshore wind, as well as the weighted average cost of capital (WACC) for these technologies, have a moderate impact. The other parameter changes have a low impact on cost. For our global analysis of GHG emissions from electrolytic hydrogen production, the capacity factor of hydrogen production (and renewables) has the highest impact (Supplementary Figure 36). Hydrogen leakage and embodied emissions of solar PV and onshore wind, as well as their component lifetimes, have a moderate impact. The other parameter changes have a low impact on GHG emissions. Hydrogen leakage has a substantial effect due to the potential (indirect) GHG impact of leaking hydrogen into the atmosphere. Embodied emissions from solar PV and onshore wind are attributed to the manufacturing processes of PV wafers and wind turbines, respectively.

Figure R4: Sensitivity analysis changing potentially influencing parameters by +10% and -10% for costs and GHG emissions.

Finally, we have updated shapefiles (using the shapefile of the World Bank³) and minor assumptions in the analysis (such as updating the Python package premise⁴ from v1.8.1 to v2.0.0 to update LCIs, e.g. for the PEM electrolyzer) to use the most recent data. We are confident that our manuscript has been further improved based on your comments and those of the editor.

Reviewer 3:

The author's response to the critical comments are highly satisfactory.

Our sincere gratitude for your constructive feedback and review of our work.

REFERENCES

1. Antonini, C. *et al.* Hydrogen production from natural gas and biomethane with carbon capture and storage - A techno-environmental analysis. *Sustainable Energy & Fuels* **4**, 2967–2986 (2020).
2. Antonini, C. *et al.* Hydrogen from wood gasification with CCS—a techno-environmental analysis of production and use as transport fuel. *Sustainable Energy & Fuels* **5**, 2602–2621 (2021).
3. *The World Bank* 2024. <https://datacatalog.worldbank.org/search/dataset/0038272/World-Bank-Official-Boundaries> (2024).
4. Sacchi, R. *et al.* PRospective EnvironMental Impact asSEment (premise): A streamlined approach to producing databases for prospective life cycle assessment using integrated assessment models. *Renewable and Sustainable Energy Reviews* **160**, 112311 (2022).

REVIEWERS' COMMENTS

Reviewer #1 (Remarks to the Author):

I recognize that the authors have made substantial efforts to further increase the transparency of the analysis. While I still stand by my original statement that the scope of the analysis is probably a bit too broad, I think that the authors have now made the best possible effort to address the specific concerns that I raised. I have no further comments.

Future hydrogen economies imply environmental trade-offs and a supply-demand mismatch

RESPONSE LETTER

Tom Terlouw^{1,2,3*}, Lorenzo Rosa⁴, Christian Bauer³, and Russell McKenna^{2,5*}

Affiliations

¹ Separation Processes Laboratory, Institute of Energy and Process Engineering, ETH Zurich, Zurich 8092, Switzerland

² Chair of Energy Systems Analysis, Institute of Energy and Process Engineering, ETH Zurich, Zurich 8092, Switzerland

³ Technology Assessment Group, Laboratory for Energy Systems Analysis, 5232 Villigen PSI, Switzerland

⁴ Department of Global Ecology, Carnegie Institution for Science, Stanford, 94035, CA, United States of America

⁵ Laboratory for Energy Systems Analysis, 5232 Villigen PSI, Switzerland

* Corresponding author: tom.terlouw@psi.ch and russell.mckenna@psi.ch.

Dear editor and reviewer of the manuscript “*Future hydrogen economies imply environmental trade-offs and a supply-demand mismatch*” with ID: **NCOMMS-23-61509B**. We thank you for your constructive comments and time. Please find our answers to your comments in black.

Reviewer 1:

I recognize that the authors have made substantial efforts to further increase the transparency of the analysis. While I still stand by my original statement that the scope of the analysis is probably a bit too broad, I think that the authors have now made the best possible effort to address the specific concerns that I raised. I have no further comments.

Our sincere gratitude for your constructive feedback and review of our work.